# SWI/SNF ATPase silenced *HLF* potentiates lung metastasis in solid cancers

Jin Zhou [1], Austin Hepperla [2,3,4], Jeremy M. Simon [2,3,4], Kangsan Kim[1], Qing Hu [1], Chuanhai Zhang[5], Lei Dong [6], Lianxin Hu [1], Cheng Zhang[1], Chengheng Liao [1], Alice Fang[1], Yayoi Adachi[1], Haoyong Fu[1], Tao Wang[1], Qian Liang[1], Fangzhou Zhao[1], Hongyi Liu[1], Masashi Takeda[1], Jun Fang[1], Hua Zhong[1], Peter Ly [1], Lu Wang [7], Payal Kapur [1,8], Lin Xu [6], Liwei Jia[1], Srinivas Malladi [1], James Brugarolas [8,9], M. Celeste Simon [10] ✉, Bo Li [11] ✉ & Qing Zhang [1,12] ✉

Metastasis is the main cause of cancer-related deaths, yet the underlying mechanisms remain elusive. Here, using clear cell renal cell carcinoma (ccRCC), a tumor type with frequent lung metastases, we conduct an in vivo genome-wide CRISPR-Cas9 screen and identify HLF as a potent suppressor of lung metastasis. *HLF* depletion enhances ccRCC cell migration and lung metastasis, whereas *HLF* overexpression abrogates these effects. In ccRCC patients, *HLF* expression is reduced at metastatic sites and associates with epigenetic silencing mediated by the SWI/SNF ATPase subunit BRG1. *HLF* levels negatively correlate with migration potential in collagen. Mechanistically, HLF regulates *LPXN* expression, modulating the integration of collagen's mechanical cues with the actin cytoskeleton through Paxillin, thereby suppressing cancer cell migration and lung metastasis. Overexpression of *HLF* or pharmacological inhibition of BRG1 reduces cell invasion across multiple cancer types. Our findings suggest that targeting the BRG1-HLF axis offers a promising therapeutic strategy for combating metastatic cancers.

More than 90% of cancer-related deaths result from tumor metastases[1], while the underlying mechanisms remain poorly understood. The lung is a common metastatic site for various cancers, including renal cancer. In clear cell renal cell carcinoma (ccRCC), the most common type of renal cancer (>80%), the lung is the primary site of metastasis (70%)[2]. The five-year survival rate for ccRCC patients with disease confined to the kidney is approximately 90%, whereas it drops to under 20% for metastatic ccRCC[3,4]. Using ccRCC as a model, we performed an in vivo genetic

[1]Department of Pathology, University of Texas Southwestern Medical Center, Dallas, TX, USA. [2]Lineberger Comprehensive Cancer Center, University of North Carolina School of Medicine, Chapel Hill, NC, USA. [3]Department of Genetics, University of North Carolina, Chapel Hill, NC, USA. [4]UNC Neuroscience Center, University of North Carolina, Chapel Hill, NC, USA. [5]Department of Physiology, University of Texas Southwestern Medical Center, Dallas, TX, USA. [6]Quantitative Biomedical Research Center, Department of Health Data Science and Biostatistics, Peter O'Donnell Jr. School of Public Health, University of Texas Southwestern Medical Center, Dallas, TX, USA. [7]Department of Biochemistry and Molecular Genetics, Feinberg School of Medicine, Northwestern University, Chicago, IL, USA. [8]Kidney Cancer Program, Simmons Comprehensive Cancer Center, University of Texas Southwestern Medical Center, Dallas, TX, USA. [9]Department of Internal Medicine, University of Texas Southwestern Medical Center, Dallas, TX, USA. [10]Abramson Family Cancer Research Institute, Department of Cell and Developmental Biology, University of Pennsylvania, Philadelphia, PA, USA. [11]Department of Biochemistry and Molecular Biology, Cancer Research Institute, School of Basic Medical Sciences, Southern Medical University, Guangzhou, PR China. [12]Harold C. Simmons Comprehensive Cancer Center, University of Texas Southwestern Medical Center, Dallas, TX, USA. ✉e-mail: celeste2@pennmedicine.upenn.edu; libo47@smu.edu.cn; Qing.Zhang@UTSouthwestern.edu

screening using genome-wide CRISPR-Cas9 editing in transplantable tumors to investigate the critical genes regulating lung metastasis.

Cancer metastasis is a complex multistep process beginning with the acquisition of traits that allow cells to overcome physical barriers, enabling their dissemination from the primary tumor to distant tissues. During this initial step, collagen, the primary component of the extracellular matrix (ECM) in the tumor microenvironment, forms a physical scaffold that provides tissue stiffness, creating barriers to tumor cell dissemination[5,6]. Through the in vivo CRISPR screening and functional validation, we identified hepatic leukemia factor (HLF) as a critical regulator that suppresses tumor cell metastasis to the lungs by modulating the interaction between cancer cells and collagen.

HLF is a member of the proline-/acid-rich and leucine zipper transcription factor (PAR bZIP) family, recognized for its circadian-dependent transcriptional regulation[7]. It is also a well-known chimeric transcription factor formed through chromosomal translocation with the transcription factor 3 (TCF3) gene (also known as E2A), driving oncogenic events in acute lymphoblastic leukemia (ALL)[8–11]. Intriguingly, this TCF3–HLF fusion protein has not been identified in solid cancers, and the role of HLF in solid tumors was not appreciated until recently, including carcinogenesis in liver, distant metastases in non-small cell lung cancer and cell stemness in ovarian cancer[12–14]. Our study reveals a important role for HLF in mediating lung metastasis in multiple solid cancers through cell-collagen interactions. Notably, we identified that the SWI/SNF ATPase subunit BRG1, a key regulator of enhancer activity in cancer cells[15], contributes to the enhancer loss of HLF, ultimately promoting cancer cell lung metastasis. Additionally, the BRG1 degrader AU-15330[15] exhibited strong inhibitory effects on cell invasion across multiple cancer types via HLF. These findings provide potential therapeutic opportunities for combating metastatic cancers.

## Results

### In vivo CRISPR screen identifies suppressors of metastasis

To identify critical regulators for lung metastasis, we applied a genome-wide CRISPR screening platform in ccRCC cell lines. Briefly, UMRC2 and A498 ccRCC cells were infected with a library virus at a multiplicity of infection (MOI) of -0.3 followed by puromycin selection and subcutaneous injection into NSG mice. After eight weeks, primary tumors and lung metastatic nodules were collected, and genomic DNA was extracted and analyzed through deep sequencing (Fig. 1a and Supplementary Fig. 1a). We sought to identify genes that, when deleted, became significantly enriched in lung metastases. Our analysis identified 577 concordant genes across both ccRCC cell lines where depletion promoted lung metastasis. Among the top ranked genes, we identified 10 genes in common across both cell lines (Fig. 1b).

Next, we performed analyses where we integrated information on each gene's expression in paired normal vs tumor tissues of ccRCC patients using our own UTSW kidney platform as well as normal vs ccRCC tumor or its transcriptomic subtype ccB (a ccRCC molecular subtype with poorer prognosis enriched with metastasis-related genes[16]) from TCGA database, and the association with disease-free survival (DFS). Genes that were suppressed in paired tumors (or in the ccB subtype) and exhibited lower expression levels associated with worse prognosis were selected for further evaluation. They included *UBE4B*, *CYCS*, *HLF*, *FOSB*, *B4GALT6*, *TRMT5* (Supplementary Fig. 1b–d). Of note, we also found that all these genes displayed lower expression in primary tumors compared to paired normal kidney from ccRCC patients with lung metastasis in most cases (Supplementary Fig. 1e). We then overexpressed these genes in 786-O ccRCC cells, except for *UBE4B* for which we were unable to achieve exogenous expression and conducted a wound healing assay and a Boyden chamber assay to examine collective cell migration and cell invasion, respectively. Among these genes, *HLF* overexpression significantly decreased cell migration and invasion, prompting us to focus on HLF (Fig. 1c, d and Supplementary Fig. 1f, g).

### HLF is downregulated in metastatic tumors

In the TCGA dataset, ccRCC from patients with higher stage (S) or grade (G) displayed lower *HLF* expression (S4 vs S1-3 or G4 vs G1-3), indicating that HLF was silenced with disease progression (Supplementary Fig. 1h). To examine the level of *HLF* in ccRCC primary tumors and their respective lung metastasis counterparts, we performed RNAScope in situ hybridization and immunohistochemistry (IHC) staining. First, we validated the *HLF* probe by examining its signal intensity for one ccRCC tumor/normal pair, confirming that *HLF* expression was significantly weaker in tumors compared to normal tissue, consistent with other mRNA analyses (Supplementary Fig. 1i); and we validated the HLF antibody by comparing its signal in 786-O cell blocks transfected with either an empty vector or HLF, confirming stronger staining in the HLF-overexpressing cells (Supplementary Fig. 1j). Notably, while there was some variability in primary ccRCC tumors, *HLF* expression at both mRNA level and protein level was lower in lung metastases (Fig. 1e).

Some ccRCCs invade intravascularly forming a tumor thrombus (TT) in the renal vein/inferior vena cava and these tumors are at enhanced risk for developing lung metastases. In previous studies, we classified TTs as TT<sup>NM</sup> (patients with no metastasis) and TT<sup>M</sup> (patients with metastasis)[17]. When comparing TT^M with TT^NM samples, *HLF* was downregulated in patient tumors with metastases (Fig. 1f). Correspondingly, TCGA data analysis revealed that lower *HLF* levels were associated with worse prognosis in ccRCC patients (Fig. 1g). In addition, expression of *HLF* was also analyzed in primary tumors and metastases in a larger patient cohort with different tumor types using the TNMplot database[18]. We found that *HLF* levels were lower in most metastatic tumors compared to primary tumors including kidney, liver, pancreas, prostate and skin cancers (Fig. 1h).

### HLF represses ccRCC lung metastasis

To assess the role of HLF in mediating ccRCC lung metastasis, both gain-of-function and loss-of-function experiments were performed in vivo and in vitro. For loss-of-function experiments, we obtained two independent *HLF* shRNAs (#3 and #4) and infected multiple ccRCC cell lines, including 786-O, Caki-1 and UMRC6. We observed that *HLF* knockdown promoted ccRCC collective migration and transwell invasion but did not grossly affect cell proliferation (Fig. 2a and Supplementary Fig. 2a–d). Lacking a reliable HLF antibody for detecting endogenous protein by western blotting, we used a CRISPR knock-in approach to add a Flag tag to the endogenous *HLF* locus in the genome (Supplementary Fig. 2e, f). Subsequently, we designed two independent sgRNAs against *HLF* and confirmed *HLF* depletion by western blot (against Flag tag) in 786-O cells. In conjunction with the sgRNAs, we also generated an exogenous V5-*HLF* that would be resistant to sgRNA depletion. In these cells we found that depletion of *HLF* enhanced cell invasion (Supplementary Fig. 2g, h) and that enhanced-migration and invasion could be dampened by expressing the sgRNA-resistant V5-*HLF* (Fig. 2b and Supplementary Fig. 2i, j). Importantly, 2D and 3D colony formation was not affected by either *HLF* knockdown or rescue (Supplementary Fig. 2k, l). Conversely, *HLF* overexpression in ccRCC cell lines decreased collective migration and transwell invasion without affecting cell proliferation (Fig. 2c and Supplementary Fig. 3a–k).

We then performed orthotopic injection of *HLF*-depleted 786-O cells in NSG mice and found that loss of *HLF* promoted lung metastasis without affecting primary tumor growth (Fig. 2d, e and Supplementary Fig. 3l, m). Conversely, *HLF* overexpression decreased the probability to develop tumors in lungs when we counted the percentage of lung tumor formations, and decreased lung metastatic tumor signal as well as circulating tumor cells (CTCs) in blood without affecting primary tumor growth (Fig. 2f–h and Supplementary Fig. 3n, o). In addition, *HLF* overexpression inhibited lung colonization completely by intravenous tail vein injection (Fig. 2i–k). Together, these indicate that HLF is both necessary and sufficient to mediate ccRCC lung metastasis.

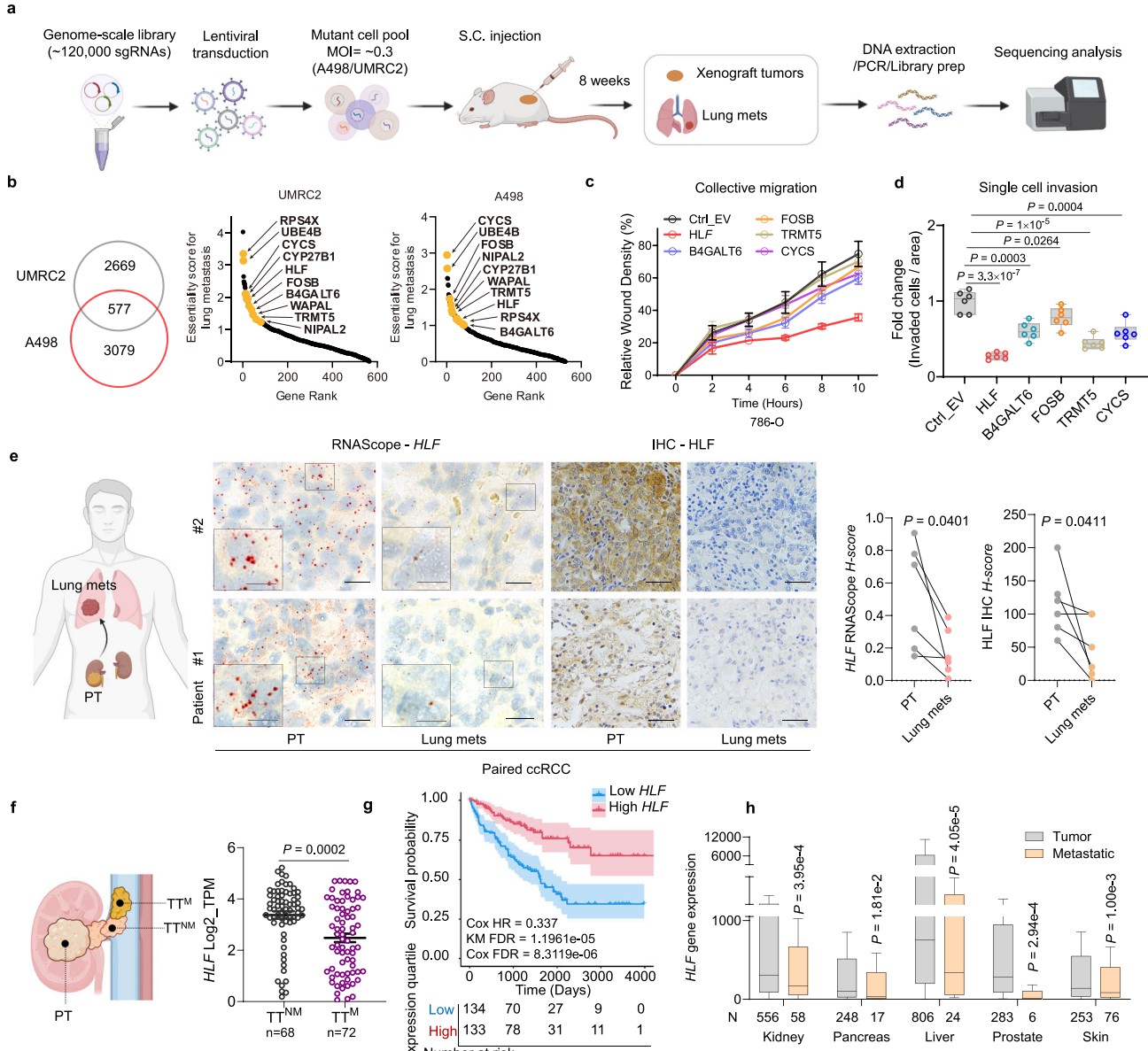

**Fig. 1 | In vivo genome-wide CRISPR-Cas9 screening identifies HLF as a lung metastasis suppressor in ccRCC. a** Schematic outline of whole-genome CRISPR screen to identify essential genes regulating tumor metastasis in xenograft mouse models for kidney cancer. Created in BioRender. *Zhou, J.* (2025) https://BioRender.com/x5scj3g. **b** Venn diagram showing the number of overlapping genes essential for UMRC2 and A498 cells metastasizing to the lung, and shared genes were ranked based on their essentiality scores for lung metastasis. The top ten genes with highest essentiality scores for both cell lines were denoted. **c, d** Quantification of wound healing assay ($n = 3$ independent cell cultures) (**c**) and transwell invasion assay ($n = 6$ independent cell cultures) (**d**) in 786-O cells transduced with pLX304-empty vector (Ctrl_EV) or target genes cloned into pLX304 backbone. **e** Schematic showing primary tumor (PT) in kidney and corresponding lung metastatic tumors (Lung mets), representative images of *HLF* RNAScope assay and HLF IHC assay in patient renal PTs and Lung mets, and corresponding *H-score*. $N = 6$ patient samples ($n = 3$ paired / $n = 3$ unpaired ccRCC tumors), scale bar, 20 μm for RNAScope (10 μm for zoomed-in

views), 50 μm for IHC. Created in BioRender. *Zhou, J.* (2025) https://BioRender.com/x5scj3g. **f** Schematic of tumor thrombus (TT) from kidney primary tumor (PT) (left) and *HLF* mRNA level in TT^NM (tumor thrombus, patients with no metastasis) compared with TT^M (patients with metastasis), $n = 68$ in TT^NM group, $n = 72$ in TT^M group. Created in BioRender. *Zhou, J.* (2025) https://BioRender.com/x5scj3g. **g** Overall survival of *HLF* from the cancer genome atlas kidney renal clear cell carcinoma (TCGA-KIRC) dataset. Sample sizes are indicated in the figure. **h** The mRNA levels of *HLF* in primary versus metastatic tumors across the indicated cancers were analyzed using the TNMplot web tool, sample sizes are indicated in the figure. Data are mean ± s.e.m. (**c, e, f**), box plots show the median and interquartile range, and whiskers show the data range (**d, h**), the error band in the Kaplan-Meier survival plot represents the confidence interval of the estimated survival probability at each time point (**g**). One-way ANOVA followed by a post hoc Dunnett-t-test (**d**), two-sided Mann-Whitney U test (**e**), unpaired two-tailed t-test (**f**) or Dunn's test following Kruskal–Wallis test (**h**). Source data are provided as a Source Data file.

## Collagen involvement in the suppressive role of HLF on cancer cell invasion

Next, we performed RNA-seq by using two complementary systems: one with *HLF* CRISPR-KO (KO) and one with *HLF* overexpression (OE) in 786-O cells. We specifically focused on those genes that were reciprocally regulated by both KO and OE samples, which yielded 1144

genes (adjusted $p ≤ 0.05$) (Fig. 3a and Supplementary Fig. 4a). To focus our efforts, we explored the genes with the highest magnitude of expression differences after changes in HLF by requiring a calculated log2 fold-change to exceed ± 0.5, yielding 258 remaining genes (Supplementary Data 1). We performed GO analysis of these genes and found that biological process pathways related to cell migration or

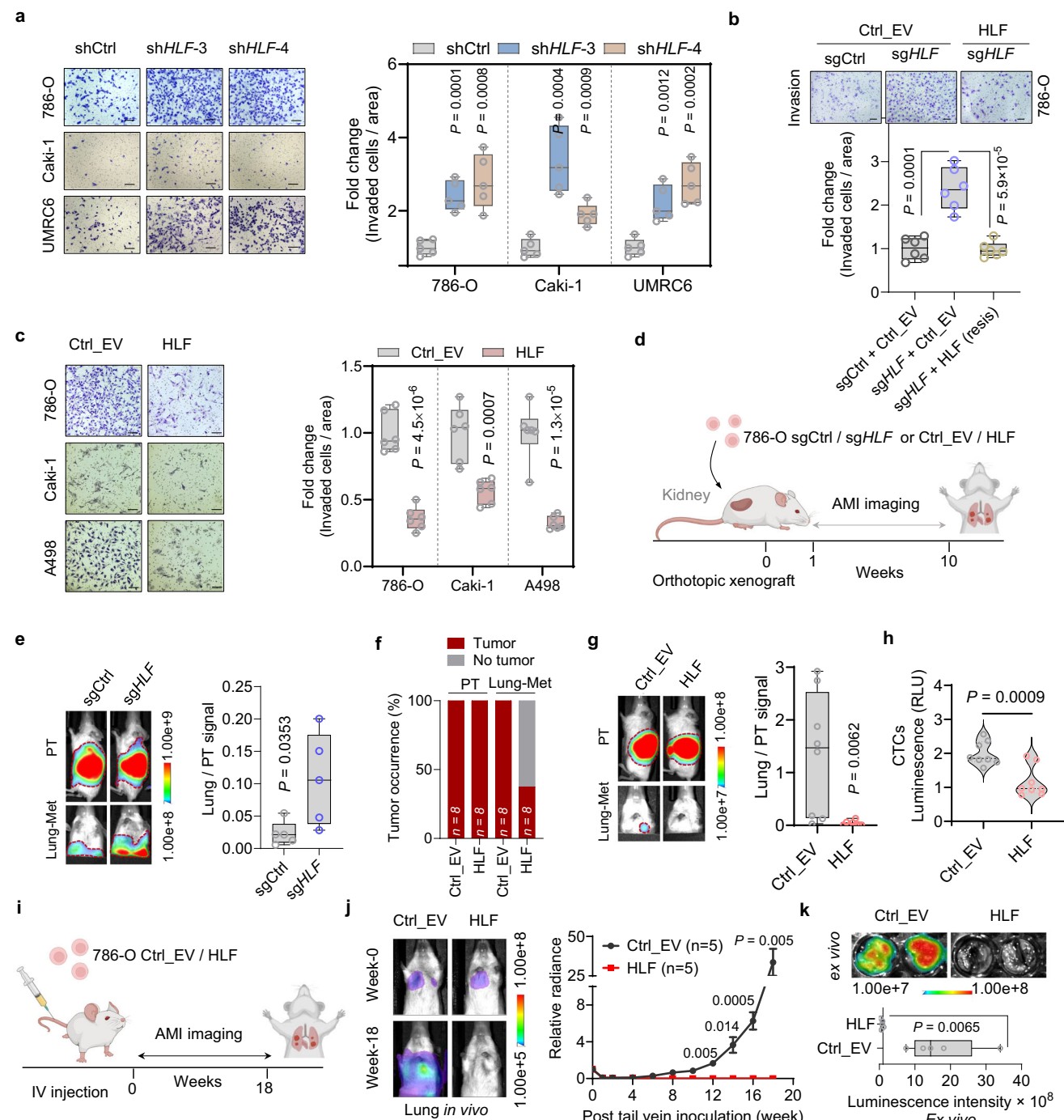

**Fig. 2 | HLF suppresses ccRCC cell invasion in vitro and lung metastasis in vivo.**
**a** Representative images and quantification of transwell invasion assay in the indicated cell lines transduced with shCtrl or *HLF* shRNAs, *n* = 6 independent cell cultures. Scale bar, 200 μm. **b** Representative images and quantification of transwell invasion assay in 786-O FLAG-KI (knock-in) cells overexpressed with empty vector or V5-sg*HLF*-resistant-*HLF* followed by infection with sgCtrl or sg*HLF*, *n* = 6 independent cell cultures. Scale bar, 100 μm. **c** Representative images and quantification of transwell invasion assay in the indicated cell lines overexpressed with empty vector (EV) or pLX304-V5-*HLF*, *n* = 6 independent cell cultures, scale bar, 100 μm. **d** Schematic of orthotopic injection into mouse kidney to observe lung metastasis. Created in BioRender. *Zhou, J.* (2025) https://BioRender.com/x5scj3g. **e** Representative images of primary tumor (PT) and metastatic tumor (Lung-met), and signals of Lung-Met relative to its matching PT derived from luciferase-labeled 786-O cells that transduced with sgCtrl or sg*HLF* followed by orthotopic injection into the renal sub-capsule of NSG mice (*n* = 5 mice in each group). **f–h** Percentage of

primary tumors (PT) and metastatic tumors (Lung-met) forming calculation (**f**), representative images of PT / Lung-met and signals of Lung-Met relative to its matching PT (**g**), and quantification of circulating tumors cells (CTCs) (**h**) analyzing from 786-O luciferase stable cell lines that transduced with empty vector (EV) or pLX304-V5-*HLF* followed by orthotopic injection into the renal sub-capsule of NSG mice (*n* = 8 mice per group). **i–k** Schematic (**i**), representative images and quantification of in vivo lung bioluminescence imaging (**j**) and representative images and corresponding quantification of lung mets by ex vivo imaging (**k**) from 786-O luciferase stable cells overexpressed with empty vector (Ctrl_EV) or pLX304-V5-*HLF* post tail vein inoculation into NSG mice, *n* = 5 mice in each group. Created in BioRender. *Zhou, J.* (2025) https://BioRender.com/x5scj3g. Box plots show the median and interquartile range, and whiskers show the data range (**a**, **b**, **c**, **e**, **g**, **k**), violin plots show the median and interquartile range (**h**). One-way ANOVA followed by a post hoc Dunnett-t-test (**a**, **b**) or unpaired two-tailed t-test (**c**, **e**, **g**, **h**, **j**, **k**). Source data are provided as a Source Data file.

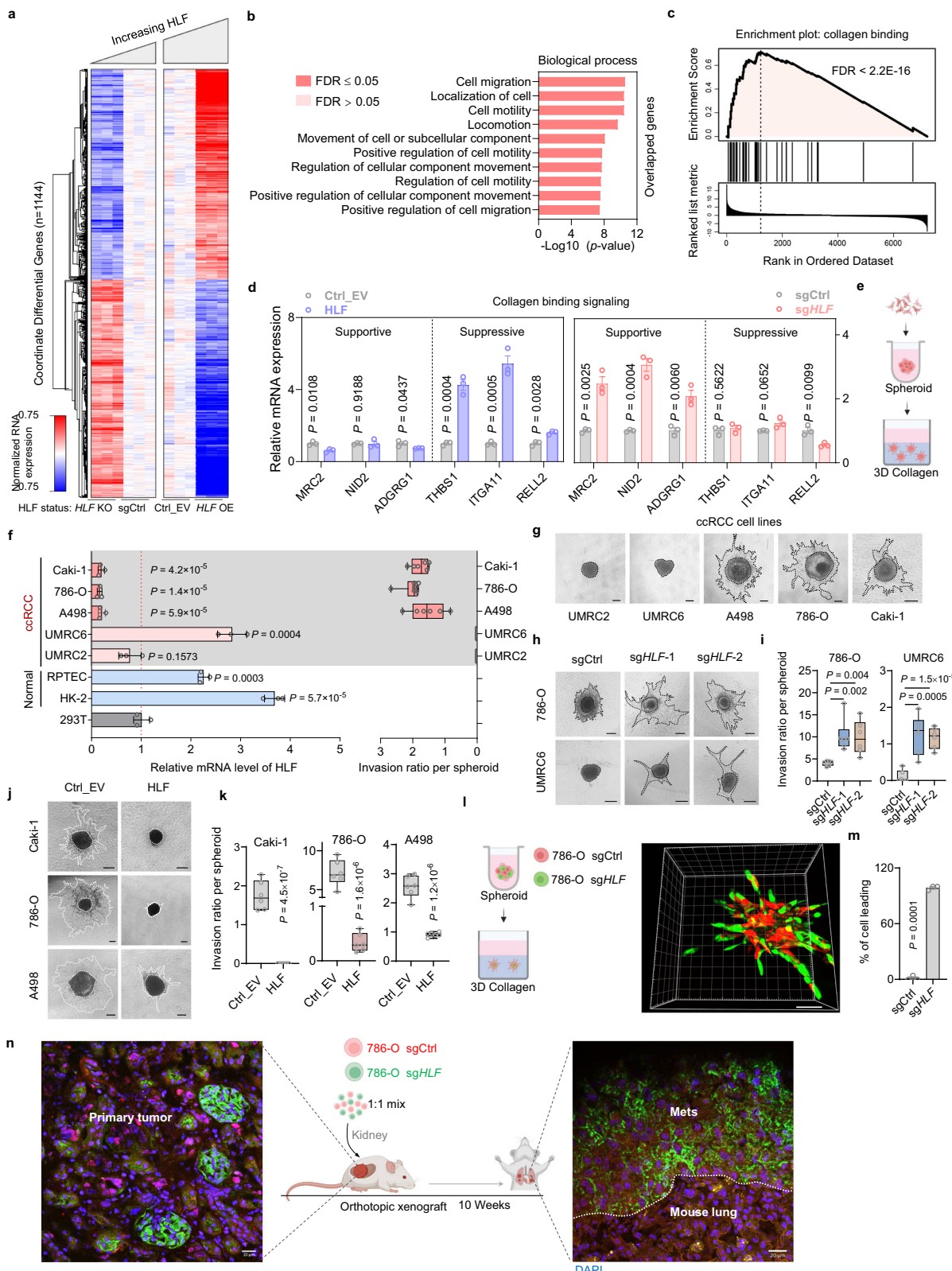

motility were enriched (Fig. 3b). More specifically, pathways that positively regulate migration were enriched among OE downregulated genes or KO upregulated genes (Supplementary Fig. 4b). In addition, GO analysis indicated enrichment in cell-cell/cell-matrix interactions, while GSEA of the OE data further specified these with "collagen binding" (Fig. 3c and Supplementary Fig. 4c, d). We performed a comprehensive literature search on the genes enriched in the

"collagen binding" signaling pathway, and quantified the expression of the genes that may affect cell migration in both HLF OE and Kcompeting

O 786-O cells. We found that HLF regulates the expression of these genes in a coordinated manner. Specifically, expression levels of genes involved in cell-matrix interaction-related supportive signaling that promotes cell migration were uniquely enhanced by HLF KO,

**Fig. 3 | Cell-collagen interaction supports the suppressive role of HLF in cell invasion. a** RNA-seq analysis of 786-O cells with HLF overexpression (OE) or knockout (KO), *n* = 3 biological replicates. **b** GO analysis of overlapped genes in both *HLF* OE and KO group. **c** GSEA analysis of the *HLF*-OE seq data in terms of the collagen binding pathway. **d** RT-qPCR of the critical genes enriched in the "collagen binding" pathway from GSEA analysis, n = 3 biological replicates. **e** Schematic showing spheroid formation followed by embedding in collagen to monitor cell invasion. Created in BioRender. *Zhou, J.* (2025) https://BioRender.com/x5scj3g. **f, g** RT-qPCR (*n* = 3 biological replicates) of *HLF* mRNA level (left) and quantification of 3D spheroid invasion (*n* = 6 independent cell cultures) (Right) (**f**), and corresponding representative images (**g**). Scale bar, 100 μm. **h, i** Representative images (**h**) and quantification (**i**) of 3D spheroid invasion in cells with *HLF* KO after 2 days (786-O) or 6 days (UMRC6) of collagen culturing, *n* = 6 independent cell cultures. Scale bar, 200 μm. **j, k** Representative images (**j**) and quantification (**k**) of 3D spheroid invasion after 3 days (786-O) or 6 days (Caki-1, A498) of collagen culturing,

*n* = 6 independent cell cultures. Scale bar, 100 μm. **l, m** Scheme of 3D co-culture in collagen followed by fluorescence imaging of the live cell spheres (**l**), and quantification (**m**) of cell leading protrusions after 3 days. (n = 3 spheres from three independent cell cultures). Scale bar, 100 μm. Created in BioRender. *Zhou, J.* (2025) https://BioRender.com/x5scj3g. **n** Scheme of the orthotopic injection of mixed sgCtrl (red) and sg*HLF* (green) cells into NSG mice. Primary tumors and lung metastases were dissected and subjected to immunofluorescence staining. Scale bar, 20 μm. Consistent results were observed across all five mice. Created in BioRender. *Zhou, J.* (2025) https://BioRender.com/x5scj3g. Data are mean ± s.e.m. (**d**, f_left), box plots show the median and interquartile range, and whiskers show the data range (f_right, **i**, **k**). DESeq2's Wald test (two-sided test) and False Discovery Rate (FDR) correction was applied (**a**), Fisher's exact test and FDR correction (**b**), One-way ANOVA followed by a post hoc Dunnett-t-test (**f**_left, I) or unpaired two-tailed t-test (d, f_right, **k**, **m**). Source data are provided as a Source Data file.

including MRC2[19], NID2[20], ADGRG1[21], whereas suppressive signaling that inhibit cell migration including THBS1[22], ITGA11[23] and RELL2[24] were specifically enhanced by HLF OE (Fig. 3d).

Cell behavior in vitro is typically examined in a two-dimensional (2D) environment, while in the body, cells exist in a three-dimensional (3D) extracellular matrix environment rich in type I collagen. Given the enrichment of cell-matrix/collagen signatures, we applied a 3D cell culture system to monitor cell migration in collagen to establish a more physiologically relevant system to model cell-matrix/collagen interaction (Fig. 3e). *HLF* was detected in both normal kidney cell lines and ccRCC cell lines by RT-qPCR, and although not all ccRCC cell lines displayed lower *HLF* expression compared with normal cells, we were intrigued by our observation that ccRCC cell lines lower in *HLF* (Caki-1, 786-O and A498) can migrate in 3D collagen compared to other ccRCC cell lines (UMRC2 and UMRC6) (Fig. 3f, g), This suggests a negative correlation between *HLF* level and migration ability in collagen.

Using our *HLF* knockout 786-O cell line, we observed enhanced ccRCC migration ability in 3D collagen after 2 days of culturing, while *HLF* knockout in UMRC6 cells, which did not exhibit migration in 3D collagen, enabled cell migration in collagen after 6 days of culturing (Fig. 3h, i and Supplementary Fig. 4e). In contrast, overexpression of *HLF* in 786-O, Caki-1 and A498 cells decreased migration in collagen (Fig. 3j, k). As solid tumors are usually highly heterogeneous, we found it prudent to investigate whether individual cells with lower *HLF* levels may be more metastatic. Cell competition is an important factor for tissue homeostasis and can result in the displacement of latent metastatic cells from the primary tumor[25]. To examine whether *HLF* loss results in a migration competitive advantage, we labeled *HLF*-depleted and control cells with GFP and RFP respectively and cultured them together at 1:1 ratio followed by monitoring. While *HLF* depletion did not result in a cell proliferation advantage (the ratio of GFP and RFP remained relatively constant during the 6-day co-culture experiment; Supplementary Fig. 4f, g), it promoted a leader cell phenotype in protrusions within the 3D collagen matrix in vitro (Fig. 3l, m) and enhanced lung metastasis in vivo (Fig. 3n). These findings suggest that *HLF* loss increases the migratory and invasive potential of ccRCC cells, thereby facilitating lung metastasis.

**LPXN mediates the function of HLF on lung metastasis in ccRCC**
To identify direct HLF target genes that may mediate the effect of HLF on cell migration and metastasis, we performed ChIP-seq with exogenously tagged HLF (HA-HLF). By integrating ChIP-seq and RNA-seq data, we focused on 134 genes associated with mobility pathways among the previously identified 258 genes with the highest expression changes (Fig. 4a). Eight of these genes were positively regulated by HLF, correlated with cell-matrix interaction and displayed strong HLF binding at their promoters or enhancers which we call HLF direct targets (Supplementary Fig. 5a). We included both enhancer and promoter regions as HLF was previously reported to affect gene

expression through epigenetic regulation of either or both[8,26]. The genes identified as direct HLF targets involved in cell migration were *CD44*, *HTRA1*, *AJAP1*, *NECTIN1*, *LPXN*, *ADAMTSL1*, *ICAM2*, and *NFATC2* (Fig. 4a). RT-qPCR was performed to confirm their gene regulation by HLF, where *HLF* depletion led to decreased expression, and *HLF* overexpression promoted their expression (Supplementary Fig. 5b). ChIP-PCR was performed to confirm their enrichment upon HLF immunoprecipitation (Supplementary Fig. 5c). Individual overexpression of these eight genes in 786-O cells (Supplementary Fig. 5d) showed that *CD44*, *AJAP1*, *LPXN* and *ICAM2* robustly decreased transwell invasion and 3D collagen invasion, indicating that they may serve as functional targets of HLF (Supplementary Fig. 5e–g). Next, to examine which target genes may mediate the role of HLF on ccRCC cell invasion, we depleted *HLF* followed by overexpression of individual target genes and examined 2D and 3D invasion phenotypes. We observed that *HLF* depletion led to increased invasion, whereas the phenotype was ameliorated by overexpression of *AJAP1* and *LPXN* (Fig. 4b and Supplementary Fig. 5h). Subsequent experiments knocking down *AJAP1* or *LPXN* alone via shRNA showed that *LPXN* knockdown, but not *AJAP1*, led to increased cell invasion (Supplementary Fig. 6a-d). This indicates that LPXN serves as a mediator of migration in either gain-of-function or loss-of-function studies of HLF. To examine the role of LPXN on ccRCC lung metastasis regulated by HLF, we also orthotopically injected sgCtrl, sg*HLF* (*HLF* depleted) or sg*HLF* cells with concurrent *LPXN* overexpression (sg*HLF* + *LPXN*) into the kidney capsule. These cells displayed comparable primary tumor growth. However, increased lung metastasis by *HLF* depletion was abrogated upon *LPXN* overexpression, suggesting that LPXN is the critical mediator for regulating ccRCC lung metastasis initiated by *HLF* loss. (Fig. 4c and Supplementary Fig. 6e).

Having identified LPXN as the critical mediator of HLF-related cell migration, we sought to elucidate how HLF regulates LPXN. We confirmed that LPXN is positively regulated by HLF at the protein level, where *HLF* overexpression led to increased LPXN and *HLF* depletion resulted in decreased LPXN across several ccRCC cell lines. Positive LPXN regulation by HLF was further validated by restoration of LPXN in *HLF*-depleted 786-O cells complemented with sgRNA-resistant exogenous tagged HLF (V5-HLF) (Fig. 4d and Supplementary Fig. 6f, g). To investigate the regulatory mechanism, we integrated our HLF ChIP-seq data (HA-HLF) with ChIP-seq data from histone H3 Lysine 4 trimethylation (H3K4me3), H3K27me3, histone H3 Lysine 27 acetylation (H3K27ac), H3K4me1 and Endothelial PAS Domain Protein 1 (EPAS1, which encodes HIF2α) in 786-O cells from previous publications[27,28] (Supplementary Fig. 6h). Sites with both H3K27ac and H3K4me1 represent active enhancers[29], whereas sites with overlapping H3K27ac and H3K4me3 demarcate active transcription at start sites[30]. Consistent with previous reports that HLF is involved in enhancer remodeling in leukemia[8], we found that many HLF binding sites also displayed H3K27ac and H3K4me1 signal (Supplementary Fig. 6h),

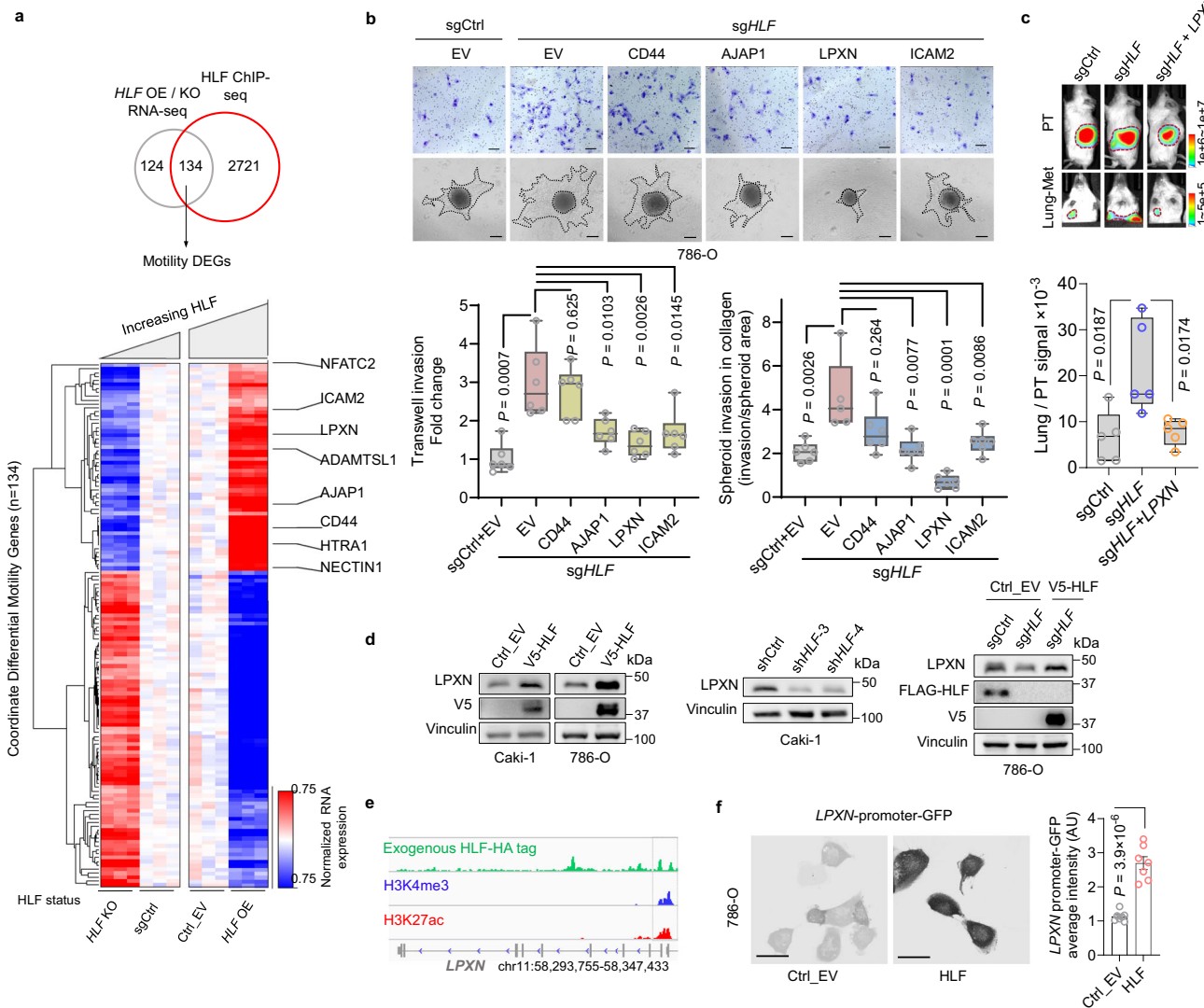

**Fig. 4 | LPXN serves as the critical effector of HLF mediating lung metastasis.**
**a** Integrated analyses of HA-tagged HLF ChIP-seq and RNA-seq data. Coordinated DEGs (1144 genes) from RNA-seq data of both *HLF* overexpression (OE) and depletion (KO) were narrowed down to 258 genes using a Log2FC cut-off of ±0.5, by overlapping with HLF top 10k peak-associated DEGs (2855 genes) from HLF-HA ChIP-seq data, and cell motility-related genes (134 genes) were identified. Potential critical target genes are indicated on the right. *n* = 3 biological replicates for RNA-seq, *n* = 2 technical replicates for ChIP-seq. **b** Representative images and quantification of transwell invasion assay and 3D collagen invasion assay in 786-O cells transduced with sgCtrl or sg*HLF* followed by infection with lentivirus expressing pLX304-empty vector (Ctrl_EV) or HLF-targeted genes in pLX304 backbone, *n* = 6 independent cell cultures. Scale bar, transwell invasion, 100 μm; 3D spheroid invasion in collagen, 200 μm. **c** Representative images of primary tumors (PT) and lung metastatic tumors (Lung-met), and signals of Lung-Met relative to its matching PT from luciferase-labeled 786-O cells transduced with sgCtrl or sg*HLF* followed by infection with lentivirus expressing pLX304-empty vector (Ctrl_EV) or pLX304-

*LPXN* and orthotopic injection into the renal sub-capsule of NSG mice, *n* = 5 mice in each group. **d** Immunoblotting analysis of the indicated ccRCC cell lines following *HLF* overexpression or knockout. Each experiment has been repeated at least two times with similar results. For the left panel, the samples derive from the same experiment but different gels for LPXN, Vinculin and another for V5 were processed in parallel. For the right panel, the samples derive from the same experiment but different gels for LPXN, Vinculin, another for FLAG and another for V5 were processed in parallel. **e** ChIP-seq binding peaks of HA-tagged HLF (exogenous), H3K4me3, and H3K27ac at the promoter region of *LPXN*. **f** *LPXN* promoter reporter assay and quantification in 786-O cells overexpressed with empty vector (Ctrl_EV) or pLX304-V5-*HLF* (HLF) followed by infection with lentivirus expressing pEZX-LvPF02-*LPXN* promoter-eGFP. 40x magnification, scale bar, 50 μm, *n* = 7 independent cell cultures. Data are mean ± s.e.m. (**f**), box plots show the median and interquartile range, and whiskers show the data range (**b**, **c**), One-way ANOVA followed by a post hoc Dunnett-t-test (**b**, **c**) or unpaired two-tailed t-test (**f**). Source data are provided as a Source Data file.

indicating that enhancer-related mechanisms may also be involved in ccRCC. Correspondingly, we observed active enhancer signal in the region of the *LPXN* gene (Supplementary Fig. 6i). To explore whether the expression of *LPXN* is regulated by HLF through enhancers, we applied the CRISPR/dCas9-based enhancer-targeting epigenetic editing system, enCRISPRi (coupling the deactivated Cas9 to repressor KRAB), which is an efficient method for enhancer perturbation[31]. We located the HA-HLF and H3K4me1 binding site at the *LPXN* gene region and designed primers to its sequence to perform enCRISPRi, but we

did not observe downregulation of *LPXN* upon enhancer disruption (Supplementary Fig. 6i, j), nor upon application of the histone deacetylase inhibitor, vorinostat (Supplementary Fig. 6k). However, HLF also overlapped H3K27ac and H3K4me3 at the promoter region of *LPXN* (Fig. 4e). Considering that HLF is a transcription factor, we proposed that HLF might regulate the expression of *LPXN* though promoter binding, and this was confirmed by *LPXN*-promoter reporter assay (Fig. 4f), where HLF overexpression induced LPXN promoter activity. Furthermore, by using a pan histone demethylase inhibitor, C70, we

observed upregulated *LPXN* mRNA and protein levels in tandem with increased H3K4me3 in cells (Supplementary Fig. 6l, m).

## HLF-LPXN-PXN coordinates cytoskeleton and collagen mechanical cues

LPXN was reported to suppress the tyrosine phosphorylation of paxillin (PXN)[32], which is crucial for actin cytoskeleton organization and mechanotransduction during adhesion, essential for cell migration[33,34]. Consistent with these reports, *HLF* depletion led to decreased LPXN levels and increased the phosphorylation of PXN at Tyrosine 118 (p-PXN-18). In contrast, *HLF* overexpression resulted in increased LPXN expression with decreased p-PXN-118 (Fig. 5a, b and Supplementary Fig. 7a-c). More importantly, the phosphomimetic version of paxillin at Tyrosine 118 (Y118E) significantly rescued HLF overexpression-induced inhibition of 3D collagen invasion, whereas the non-phosphorylatable version (Y118F) showed a much weaker rescue effect (Fig. 5c and Supplementary Fig. 7d).

Cancer cells' ability to mechanically adjust to extracellular matrix stiffness was reported to correlate with their invasive potential which may aid colonization at physically different secondary sites in the body[6]. Collagen is the main component of the extracellular matrix (ECM) which contributes to its stiffness, and collagen type I is the most abundant form of collagen that builds a physical scaffold providing stiffness to the tissue[6]. We applied a single cell-tracking assay in a 2D stiff collagen model[35], and found that collagen is involved in the function of HLF on cell migration. Briefly, *HLF* depletion enhanced cell migration (increased velocity, Euclidean distance and directionality) on a stiff collagen surface, while no differences were observed on surfaces without stiff collagen coating. Notably, the increased single cell migration on the stiff collagen surface due to *HLF*-loss can be rescued by exogenous *LPXN* expression (Fig.5d–f and Supplementary Fig. 7e–i). Additionally, *HLF* overexpression reduced single-cell migration on a stiff collagen surface, as evidenced by decreased cell velocity and a shorter Euclidean distance between the initial and final points. Importantly, no differences in cell velocity or Euclidean distance were observed on surfaces without stiff collagen coating, although increased directionality was noted (Fig. 5g–i and Supplementary Fig. 7j).

Cells sense stiffness primarily through adhesions that couple the extracellular matrix of interacting cells to the actin cytoskeleton, with PXN playing a crucial role during this process[35,36]. Consistently, on a stiff collagen surface, the knockdown of *LPXN* increased the organization of actin filaments which was reflected by increased F-actin intensity and filament number (Supplementary Fig. 7k–n). We also observed enhanced F-actin organization following *HLF* knockout, which could be attenuated by *LPXN* overexpression (Fig. 5j–l). Together, *HLF* loss decreases LPXN expression, enhancing the integration of collagen stiffness and the actin cytoskeleton through paxillin, ultimately promoting cancer cell migration (Fig. 5m).

## HLF is epigenetically silenced in metastatic ccRCC

To further study the role of HLF in metastases, we focused on M1A, a metastatic subpopulation generated from 786-O cells by tail vein injection[37]. We found that M1A cells exhibited lower expression of *HLF* coupled with higher invasion ability in collagen compared with paired parental cells (Supplementary Fig. 8a, b). Overexpression of *HLF* in M1A cells decreased collective cell migration, transwell invasion, as well as 3D collagen invasion (Supplementary Fig. 8c–f). However, this tail vein metastatic model involves direct injection of cancer cells into the circulatory stream bypassing critical steps in the multi-step process of invasion and metastasis. To evaluate in a physiologically relevant setting, we generated a primary tumor cell line (786-O-P, from primary tumor) and a metastatic tumor cell line (786-O-LM, from lung metastasis) from 786-O cells following orthotopic injection into the kidney of mice (Fig. 6a). We confirmed lower expression of *HLF* and its target

genes in the metastatic derivatives by RT-qPCR as well as lower LPXN protein levels (Fig. 6b, c and Supplementary Fig. 8g). Furthermore, decreased *HLF* mRNA level was associated with increased transwell invasion and migration in 3D collagen, while no difference in cell proliferation was observed (Fig. 6d and Supplementary Fig. 8h–j). When the cells were expanded in culture and reinoculated into mice by tail vein injection, the 786-O-LM cells exhibited higher lung metastatic activity compared with the parental population (Fig. 6e and Supplementary Fig. 8k). Restoration of *HLF* or *LPXN* in 786-O-LM cells decreased cell invasion and 3D collagen migration in vitro (Supplementary Fig. 8l–o). Furthermore, restoration of *HLF* in 786-O-LM cells inhibited lung metastasis in vivo (Supplementary Fig. 8p–u).

*HLF* expression gradually decreased in 786-O cells after injection into kidney and following metastasis to the lungs (786-O-LM < 786-O-P <parental 786-O) (Supplementary Fig. 9a), however the molecular mechanism behind this silencing remains to be determined. Mechanistically, we found little evidence for hypoxia, altered mRNA stability or increased DNA methylation as the explanation for decreased expression of HLF (Supplementary Fig. 9b–f). To explore whether epigenetic changes in transcriptional accessibility could explain our observation, we performed the assay for transposase-accessible chromatin with sequencing (ATAC-seq) in paired 786-O-P and 786-O-LM cells to determine chromatin accessibility differences across the genome. The locus that contains the *HLF* gene showed reduced accessibility in 786-O-LM cells, where we also observed decreased H3K4me1 and H3K27ac in metastatic M1A cells compared to parental cells (Fig. 6f). Consistent with this epigenetic regulation, treatment of the metastatic cell line M1A with the histone deacetylase inhibitor vorinostat, either alone or in combination with the histone demethylase inhibitor C70, resulted in increased HLF mRNA and protein levels (Fig. 6g, Supplementary Fig. 9g), indicating the involvement of enhancer regulation mechanisms. To verify that *HLF* is enhancer-regulated, we applied the CRISPR/dCas9-based enhancer-targeting epigenetic editing systems, enCRISPRa (coupling the deactivated Cas9 to activator p300) and enCRISPRi (coupling the deactivated Cas9 to repressor KRAB), which are efficient methods for enhancer perturbation[31] (Fig. 6h). Sequence-specific sgRNAs were used to target the enhancer region that HLF binds to, and we found that dCas9-KRAB repressed *HLF* expression, whereas dCas9-p300 significantly activated *HLF* expression in multiple ccRCC cell lines (Fig. 6i).

## BRG1 mediates the epigenetic loss of *HLF*

Transcriptional activators or repressors can bind to enhancers to promote or block transcription, so we explored potential DNA binding proteins from public ChIP-seq data in 786-O cells using the ChIP-Atlas platform[38]. The proteins that were found to bind to the two identified enhancers regions, and thus may serve as potential *HLF* regulators, included SWI/SNF ATPase subunit BRG1 (Brahma-related gene 1), which can be an activator or repressor, as well as ARNT (Aryl Hydrocarbon Receptor Nuclear Translocator) and EP300 (E1A Binding Protein P300), both of which usually serve as activators (Fig. 6j). By performing gene ablation experiments using sgRNAs (or shRNAs), depletion of *ARNT* or *EP300* did not cause a consistent change in *HLF* expression (Supplementary Fig. 9h–j). Conversely, *SMARCA4* (encoding BRG1) depletion led to increased expression of *HLF* while *SMARCA4* overexpression resulted in decreased *HLF* expression, suggesting that BRG1 may act as a transcriptional repressor of *HLF* (Fig. 6k and Supplementary Fig. 9k–m). In addition, a VHL-based proteolysis-targeting chimera (PROTAC) degrader of BRG1, called AU-15330[15], was used to treat two non-ccRCC cancer cell lines that express VHL (A375 MA2, HCT116) as well as a normal renal epithelial cell line (HK-2). We observed decreased BRG1 levels corresponding to increased expression of *HLF* in the cancer lines, whereas a very minor effect was observed in normal cells (Fig. 6l–n), suggesting that BRG1-HLF regulation may be more specific to cancer cells. BRG1 levels were elevated

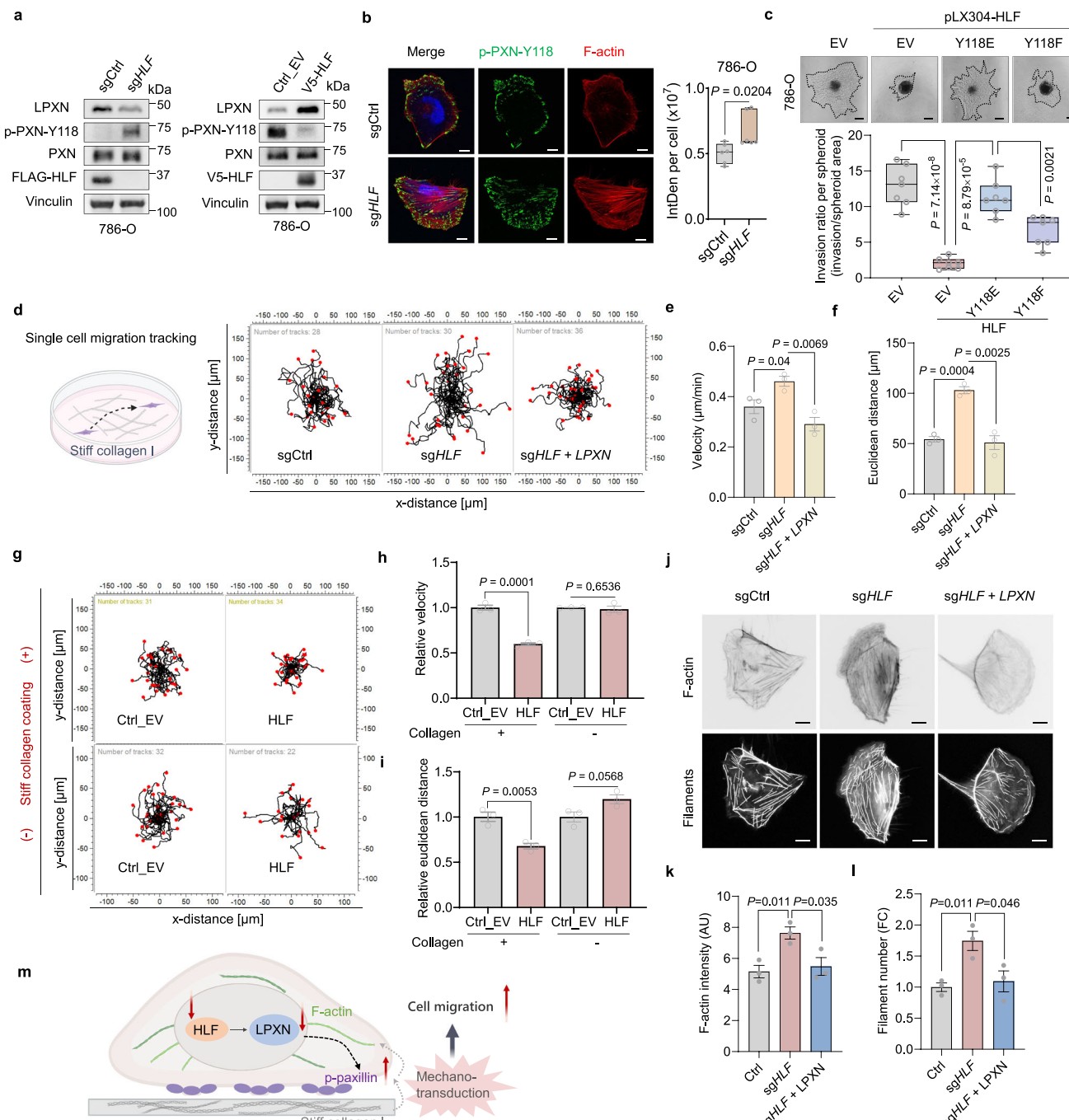

**Fig. 5 | HLF regulates cell migration via the LPXN-PXN axis. a** Immunoblotting analysis of 786-O cells following *HLF* knockout or overexpression. Each experiment has been repeated at least two times with similar results. For the left panel, the samples derive from the same experiment but different gels for LPXN, another for p-PXN-Tyr118, another for PXN and another for FLAG, vinculin. For the right panel, the samples derive from the same experiment but different gels for LPXN, Vinculin, another for p-PXN-Tyr118, another for PXN and another for V5 were processed in parallel. **b** Representative fluorescence images in cells following *HLF* knockout and cultured on stiff surface (50 μg/ml collagen coating). Nucleus were stained blue, n = 6 independent cell cultures, scale bar 10 μm. **c** Representative images and quantification (*n* = 7 independent cell cultures) of 3D collagen invasion assay in 786-O cells with *HLF* overexpression followed by infection with Paxillin Y118E (phosphomimetic version) or Y118F (non-phosphorylatable version). Scale bar, 200 μm. **d–f** Schematic showing single cells migrating on surface coating with stiff collagen I (50 μg/ml) and representative wind-rose plots showing cell tracks (**d**), migration velocity (**e**), and Euclidean distance (**f**) of the tracked 786-O cells cultured on stiff

collagen surface, cells from three independent cell cultures were analyzed. Created in BioRender. *Zhou, J.* (2025) https://BioRender.com/x5scj3g. **g-i** Representative wind-rose plots showing cell tracks (**g**), migration velocity (**h**) and Euclidean distance (**i**) of the tracked 786-O cells overexpressed with *HLF* and cultured on surface with or without stiff collagen coating. The analyzed cells were from three independent cell cultures. **j–l** Representative fluorescence images of F-actin and filaments (**j**), intensity of F-actin (**k**) and filament number per cell (**l**) in 786-O cells with *HLF* knockout following by *LPXN* overexpression and cultured on stiff surface. Scale bar, 10 μm. Cells from three independent cell cultures were analyzed. **m** Schematic showing the functional mechanism by which *HLF* loss decreases *LPXN* expression, enhancing the integration of collagen stiffness and the actin cytoskeleton via paxillin, ultimately potentiating cancer cell migration. Created in BioRender. *Zhou, J.* (2025) https://BioRender.com/x5scj3g. Data are mean ± s.e.m. (**e**, **f**, **h**, **i**, **k**, **l**), box plots show the median and interquartile range, and whiskers show the data range (**b**, **c**). One-way ANOVA followed by a post hoc Dunnett-t-test (**c**, **e**, **f**, **h**, **l**, **k**, **l**) or unpaired two-tailed t-test (**b**). Source data are provided as a Source Data file.

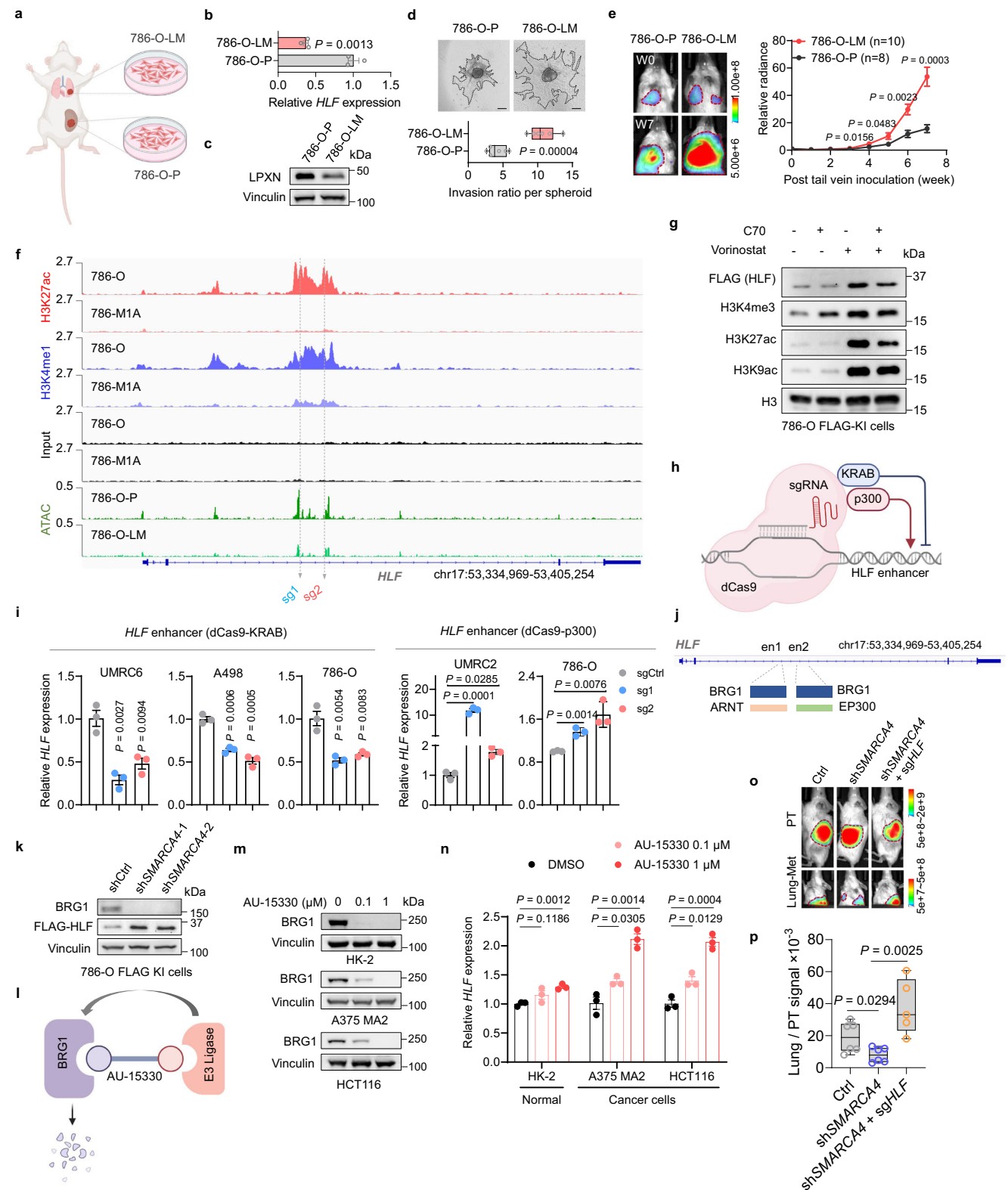

in ccRCC tumor samples compared to normal tissues (Supplementary Fig. 9n, o). Due to the lack of a reliable HLF antibody, we analyzed *HLF* mRNA levels in these paired tissues and observed a decrease in *HLF* expression in ccRCC tumor samples (Supplementary Fig. 9p). Furthermore, Pearson's correlation analysis revealed a negative correlation between BRG1 and *HLF* levels (r = −0.5844, *p* = 0.0068, Supplementary Fig. 9q). Importantly, our orthotopic injection experiment revealed that *SMARCA4* knockdown elevated HLF levels and

reduced lung metastasis, while further *HLF* knockout led to a substantial loss of HLF and a significant increase in lung metastasis (Fig. 6o, p, Supplementary Fig. 9r–t).

## HLF suppresses metastasis across multiple cancer types

To clinically correlate the expression level of *HLF* and lung metastatic potential, we established PDX cell line models from individual ccRCC patients[39] as described previously (Fig. 7a). Among the three PDX cell

**Fig. 6 | BRG1 mediates the epigenetic loss of *HLF*. a** Schematic showing the generation of primary tumor cells (786-O-P) and lung metastatic cells (786-O-LM). Created in BioRender. *Zhou, J.* (2025) https://BioRender.com/x5scj3g. **b**–**d** RT-qPCR (*n* = 3 biological replicates) (**b**), immunoblotting (**c**) (repeated at least two times independently with similar results), representative images and quantification of 3D spheroid invasion (**d**), *n* = 6 independent cell cultures. Scale bar, 200 μm.
**e** Representative images and quantification of bioluminescence imaging post tail vein inoculation into NSG mice (n = 8 mice for 786-O-P, *n* = 10 mice for 786-O-LM).
**f** ChIP-seq data of H3K27ac and H3K4me1 and ATAC-seq data at chr17:53,334,969-53,405,254 (hg19). The targeting sites of sg*HLF*s (used in **i**) at the enhancer region are indicated. **g** Immunoblotting in cells treated with C70 (20 μM) or vorinostat (2 μM) for 24 h. This experiment has been repeated three times with similar results. The samples derive from the same experiment but different gels for each target protein were processed in parallel. **h** Schematic showing CRISPR interference system (dCas9-KRAB) and CRISPR activation system (dCas9-p300). Created in BioRender. *Zhou, J.* (2025) https://BioRender.com/x5scj3g. **i** RT-qPCR in cells applied with dCas9-KRAB system (left) and dCas9-p300 system (right) respectively targeting specific *HLF* enhancer region, *n* = 3 biological replicates. **j** Potential

regulators binding at the validated regulatory enhancer region of *HLF*.
**k** Immunoblotting in cells transduced with shCtrl or *SMARCA4* shRNAs. This experiment has been repeated three times with similar results. The samples derive from the same experiment but different gels for BRG1, Vinculin and another for FLAG were processed in parallel. **l** Schematic showing the functional mechanism of AU-15330 that can degrade BRG1. Created in BioRender. *Zhou, J.* (2025) https://BioRender.com/x5scj3g. **m**, **n** Immunoblotting (**m**) and RT-qPCR (n = 3 biological replicates) (**n**) in cells treated with AU-15330 (0.1 μM, 1 μM) for 48 h. Representative immunoblots shown in **m** were repeated at least two times independently with similar results. For A375 MA2 and HCT116 cells, the samples derive from the same experiment but different gels for BRG1 and Vinculin were processed in parallel.
**o, p** Representative images (**o**) of primary tumors (PT) and lung metastatic tumors (Lung-met), and signals of Lung-Met relative to its matching PT (**p**) from orthotopic injection in NSG mice, *n* = 6 mice in Ctrl and sh*SMARCA4* group, *n* = 5 mice in rescue group. Data are mean ± s.e.m. (**b, e, i, n**), box plots show the median and interquartile range, and whiskers show the data range (**d, p**). One-way ANOVA followed by a post hoc Dunnett-t-test (**i, n, p**) or unpaired two-tailed t-test (**b, d, e**). Source data are provided as a Source Data file.

---

line models, XP374 displayed the lowest expression level of *HLF* corresponding with more invasive ability in 3D collagen compared to the other two PDX lines (XP258, XP289) (Fig. 7a and Supplementary Fig. 10a). XP258 and XP374 cells, which showed comparable growth rates in vitro (Supplementary Fig. 10b), were orthotopically injected into the kidney, and XP374 cells demonstrated stronger lung metastasis compared with XP258 cells, despite displaying comparable primary tumor growth (Fig. 7b and Supplementary Fig. 10c). Moreover, *HLF* overexpression in XP374$^{HLF\ low}$ cells decreased 3D invasion in collagen, whereas *HLF* KO in XP258$^{HLF\ high}$ cells further increased 3D invasion in collagen as well as lung metastasis in vivo (Fig. 7c, d and Supplementary Fig. 10d, e). In accordance with these findings, XP374$^{HLF\ low}$ cells displayed higher PXN activity marked by increased p-PXN-Y118 compared to XP258$^{HLF\ high}$ cells (Supplementary Fig. 10f). In addition, overexpression of HLF in XP374 cells led to decreased PXN activity (Supplementary Fig. 10g). To determine if HLF suppresses lung metastasis in other types of cancer, we evaluated multiple cancer cell lines from cancers typically associated with lung metastasis in patients, including osteosarcoma, colorectal cancer, lung cancer and melanoma. We observed that *HLF* overexpression inhibited 3D collagen invasion or transwell invasion in these cell lines in vitro (Fig. 7e, f and Supplementary Fig. 10h), as well as decreased lung metastasis of colorectal carcinoma cell line HCT116 in vivo (Fig. 7g and Supplementary Fig. 10i). Interestingly, in the lung cancer cell line H1299, *HLF* overexpression affected both primary tumor growth and lung metastasis (Fig. 7g and Supplementary Fig. 10j, k).

Additionally, the upstream regulator of *HLF*, *SMARCA4*, exhibited higher expression in metastatic tumors compared to primary tumors across multiple cancer types (Fig. 7h). The BRG1 degrader, AU-15330, demonstrated inhibitory effects on cell invasion across various cancer types, including skin cancer, ccRCC, and colon cancer, without impacting cell proliferation (Fig. 7i and Supplementary Fig. 10l, m). Notably, this inhibitory effect was diminished by HLF depletion in cells, suggesting that AU-15330 inhibits metastasis through HLF (Fig. 7j and Supplementary Fig. 10n).

## Discussion

The treatment of metastatic cancer remains a significant challenge, highlighting the necessity to identify factors influencing its progression, which could inform therapeutic approaches. Our research has identified HLF as a suppressor of metastasis, not only in renal cancer but also in other cancer types including osteosarcoma, melanoma, colorectal cancer and lung cancer. Notably, HLF functions as an oncogenic chimeric transcription factor constituting central drivers in hematological malignancies yet acts as a metastasis-suppressive

transcription factor constituting central suppressors in solid tumors. We identified that HLF suppresses cell migration and invasion through cell-collagen or cell-matrix interaction. As we know, cell-matrix interactions differ significantly in hematological and solid cancers due to the distinct environments and behaviors of these cancer types (cell adhesion can shield blood cancer stem cells from chemotherapeutic agents, promoting survival and proliferation, while in solid tumors, cell adhesion usually mediates migration and metastasis). Our findings highlight the unique role of HLF in solid tumor metastasis through cell-matrix interactions, suggesting potential therapeutic strategies for targeting HLF in metastatic cancer.

Cancer metastasis is a complex, multi-step process where *HLF* loss can influence both cancer cell dissemination from the primary tumor and localization at distant sites. Both steps involve crucial cell-matrix interactions. Our research shows that HLF decreases cancer cell migration from the primary tumor (as shown by orthotopic injection) and reduces lung localization (as evidenced by tail vein injection). The ECM, a major component of the tumor microenvironment, provides biochemical and biophysical cues that regulate tumor cell adhesion and migration. We discovered that HLF modulates a set of genes associated with cell-matrix interactions, including cell membrane receptors (MRC2, ASGRG1) and membrane proteins (NID2, THBS1, ITGA11, RELL2), which interact with ECM components and ECM-bound factors to coordinate cell adhesion and migration. This finding explains the intriguing observation that *HLF* overexpression decreases sheet migration or invasion (where cell-cell and cell-matrix interactions are involved) but does not affect the chemotactic migration of single cells.

Collagen, the most abundant ECM constituent, contributes significantly to ECM function and stiffness, both of which are implicated in tumor progression and metastasis. Using a stiff collagen coating model[35], we found that the suppressive role of HLF in cell migration is specific to stiff collagen compared to Poly-L-Lysine coating. Previous studies have shown that physical force is involved in cancer cells forming invasive protrusions in 3D cultures[40], and that F-actin is crucial for cell protrusion in a 3D matrix[41]. We observed that the F-actin organization adapts to cell movement on stiff collagen surfaces when *HLF* is manipulated, likely through LPXN. LPXN regulates PXN activity, linking the ECM to the actin cytoskeleton, and affecting actin dynamics to enable cell adhesion and migration. In summary, HLF directly (or indirectly) regulates the expression of genes involved in cell-matrix interactions. This regulation, coordinated with dynamic cell adhesion through the HLF-LPXN-PXN axis, ultimately impacts cancer cell migration. This highlights the multifaceted role of HLF in the metastatic process and suggests potential therapeutic targets for cancer treatment.

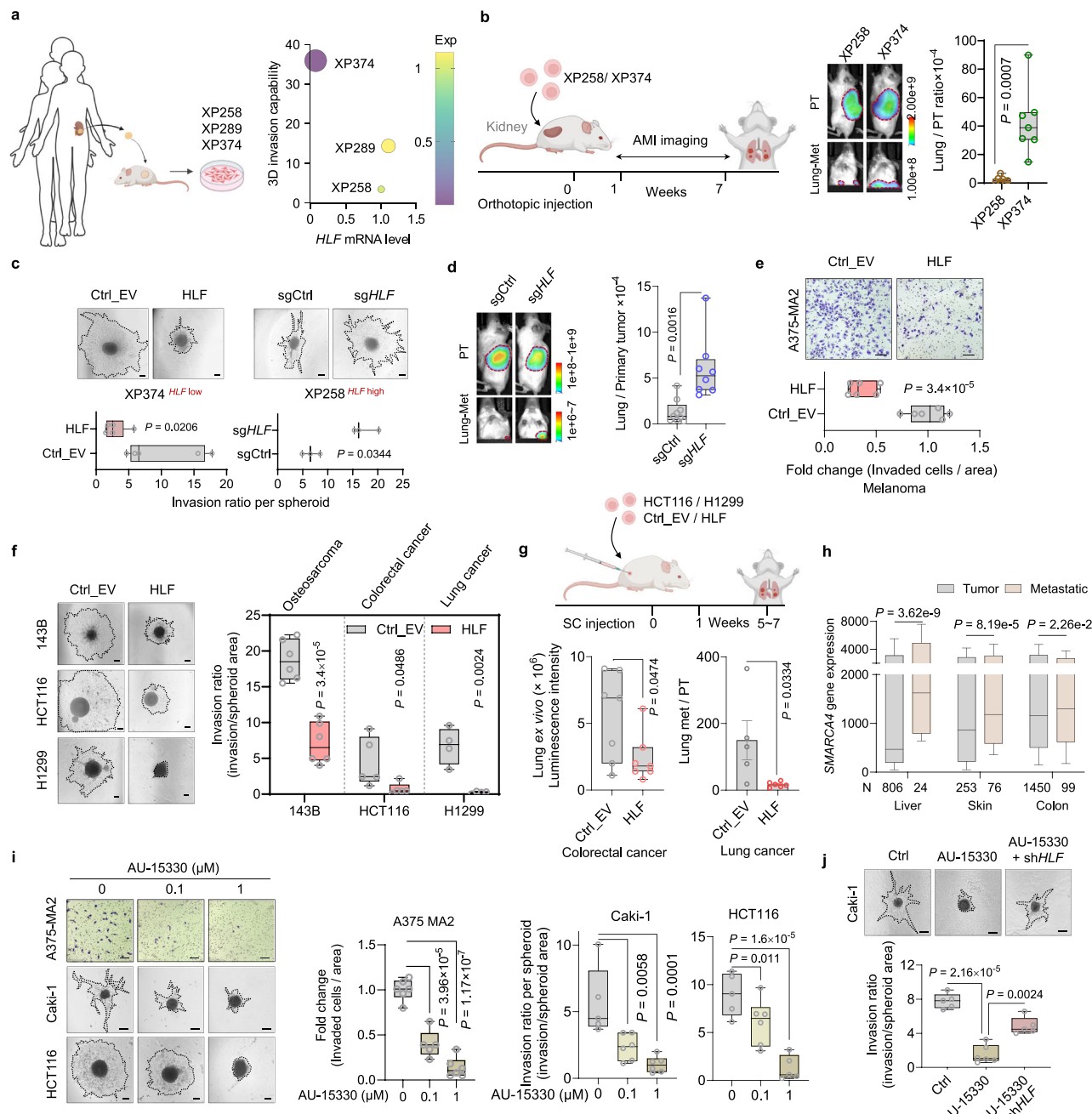

**Fig. 7 | Targeting BRG1-mediated *HLF* loss suppresses metastasis across multiple cancer types. a** Workflow for the generation of PDX cell lines, projection plot showing *HLF* mRNA level and 3D spheroid invasion capability. XP (xenograft patient), the number indicates patient ID. Created in BioRender. *Zhou, J.* (2025) https://BioRender.com/x5scj3g. **b** Schematic of kidney orthotopic implantation, representative images of primary tumors (PT) and lung metastatic tumors (Lung-met), and ex vivo signals of Lung-Met relative to its matching PT, *n* = 7 mice. Created in BioRender. *Zhou, J.* (2025) https://BioRender.com/x5scj3g. **c** Representative images and quantification of 3D spheroid invasion in XP374 cells overexpressed with HLF or XP258 cells with HLF knockout. *N* = 5 (XP374) or *n* = 3 (XP258) independent cell cultures. Scale bar, 200 μm. **d** Representative images of primary tumors (PT) and lung metastatic tumors (Lung-met), and ex vivo signals of Lung-Met relative to matched PTs from XP258 cells after orthotopic injection (*n* = 9, sgCtrl; *n* = 8, sg*HLF*). **e** Representative images and quantification of transwell invasion assay in A375-MA2 cells with *HLF* overexpression, *n* = 6 independent cell cultures, Scale bar, 100 μm. **f** Representative images and quantification of 3D spheroid invasion in cell lines overexpressed with *HLF* after 2 days (143B, *n* = 6), 3 days (HCT116, *n* = 5) and 4 days (H1299, *n* = 4) of collagen culturing from

independent cell cultures. Scale bar, 200 μm. **g** Schematic of subcutaneous implantation, quantification of luminescence signal in lungs by ex vivo imaging in HCT116 cells (n = 7 mice), and relative quantification of luminescence signal in lungs by ex vivo imaging to primary tumor weight in the H1299 cells (*n* = 5, Ctrl_EV; *n* = 6, HLF) overexpressed with *HLF* followed by subcutaneous implantation in NSG mice. Created in BioRender. *Zhou, J.* (2025) https://BioRender.com/x5scj3g. **h** The mRNA levels of *SMARCA4* in primary versus metastatic tumors across the indicated cancers. **i** Representative images and quantification of transwell invasion and 3D collagen invasion in the indicated cell lines treated with AU-15330 for 24 - 48 h, *n* = 5 or 6 independent cell cultures. Transwell invasion, 100 μm, collagen invasion, 200 μm. **j** Representative images and quantification of 3D collagen invasion in Caki-1 cells with *HLF* knockout and treatment with 1 μM AU-15330 for 48 h, *n* = 5 or 6 independent cell cultures. Scale bar, 200 μm. Data are mean (**a**, fill color shows *HLF* level, size shows invasion ability), box plots show the median and interquartile range, and whiskers show the data range. Unpaired two-tailed t-test (**b**–**g**), Dunn's test following Kruskal–Wallis test (**h**) or One-way ANOVA followed by a post hoc Dunnett-t-test (**i, j**). Source data are provided as a Source Data file.

While the formation of primary tumors is driven by a combination of genetic and epigenetic events[42], studies suggest that epigenetic regulatory mechanisms, rather than genomic alterations, play a crucial role in the metastatic cascade[43–46]. In ccRCC, metastasis has been linked to an epigenetically expanded output of the tumor-initiating pathway[37], our research reveals that HLF, which mediates metastasis suppression in our study, is regulated through enhancer mechanisms. Through epigenomic profiling, researchers identified substantial differences in enhancer activity between primary and metastatic human tumors and confirmed that positively selected enhancer elements contribute to the metastatic phenotype[47,48]. Interestingly, although they identified more enhancer loss than gain in metastatic tumors compared to primary tumors, the role of enhancer loss in mediating metastasis has been largely overlooked. HLF is recognized as an output regulator of the circadian rhythm, while circadian clocks can be controlled by epigenetics, suggesting that its expression may be retained through epigenetic mechanisms like enhancers[49]. Our study demonstrates that BRG1-induced enhancer loss leads to decreased *HLF* gene expression, promoting metastasis in ccRCC. We provide direct evidence that decreased *HLF* expression, caused by enhancer loss in metastatic kidney cancer cells, has functional consequences and directly contributes to the metastatic phenotype. This finding underscores the importance of enhancer loss as a key driver in the metastatic process and highlights HLF as a potential therapeutic target for ccRCC. In addition, a BRG1 inhibitor, FHD-286 is currently in Phase 1 clinical trials for treating metastatic uveal melanoma and advanced hematologic malignancies[50,51], highlighting the importance of BRG1 inhibition in cancer progression and metastasis. Additionally, the PROTAC degrader of BRG1, AU-15330, has shown significant efficacy in enhancer-addicted prostate cancer, and in our study, it effectively inhibited cell invasion. This underscores the connection between BRG1 and enhancer regulation, further supporting the therapeutic potential of targeting BRG1 in cancer metastasis, particularly through its regulation of HLF via enhancers.

In addition, given that HLF is a transcription factor, it likely regulates multiple genes and signaling pathways. We acknowledge that additional pathways may also play a role in HLF-mediated metastasis regulation in ccRCC and that HLF signaling pathways vary across different cancer types. For example, tensin1 (TNS1), a focal adhesion protein that regulates the molecular linkage between the extracellular matrix and the cytoskeletal network to control cell migration, has been reported to mediate HLF-driven metastasis suppression in prostate cancer[52]. However, in ccRCC, we found that HLF negatively regulates tensin1 expression, highlighting the cancer type-specific differences in HLF signaling. Additionally, previous studies have shown that HLF downregulation promotes multi-organ distant metastases in non-small cell lung cancer via NF-κB/p65 signaling[13]. However, in our study, we observed no significant changes in NF-κB/p65 signaling upon HLF regulation. Moreover, while HLF has been reported to transactivate C-Jun to promote hepatocellular carcinoma[12], and C-Jun itself has been implicated in promoting metastasis[53,54], we found that in ccRCC, C-Jun is positively regulated by HLF, which acts as a tumor suppressor in our study. Whether C-Jun plays a role in metastasis regulation in ccRCC remains unclear and warrants further investigation.

## Methods

### Ethical approval
All animal experiments were conducted in accordance with the National Institutes of Health (NIH) guidelines and were approved by the Institutional Animal Care and Use Committee (IACUC) in the University of Texas Southwestern Medical Center (Protocol # 2019-102794). The human tissues used for qPCR, RNAScope, and IHC assays were reviewed by the UT Southwestern Human Research Protection Program (HRPP), which determined that the analysis does not qualify as human subject research under 45 CFR 46.102 and therefore does not require IRB approval or oversight.

### Cell culture
786-O, A498, 293 T, A375-MA2, H1299 and Caki-1 were purchased from the American Type Culture Collection (ATCC), UMRC2 and UMRC6 were purchased from Sigma-Aldrich, HK-2 and RPTEC was acquired from Peter Ly lab at UTSW, 143B was acquired from Tao Yue lab at UTSW, HepG2 was acquired from Lu Wang lab at Northwestern University. The cell lines used in this study were cultured in Dulbecco's Modified Eagle's Medium (DMEM, Gibco) high glucose medium supplemented with 10% fetal bovine serum (FBS) and 1% penicillin/streptomycin unless specifically indicated. The normal kidney cell line HK-2 was grown in keratinocyte serum-free medium (K-SFM, Gibco). The PDX cell lines, acquired from Dr. James Brugarolas's lab at UT Southwestern, were grown in DMEM high glucose supplemented with 10% FBS, 1% Pen/Strep, 1X MEM Non-essential amino acids (Gibco, 11140050), 0.01 μg/ml EGF (Bio-Techne, 236-EG), and 0.4 μg/ml Hydrocortisone (STEMCELL, 07925). All cells were cultured in a humidified incubator at 37 °C with 5% $CO_2$. All cells were tested negative for mycoplasma by using the mycoplasma detection kit (Lonza, LT07–218) or mycoplasma elimination reagent-plasmocin (Invivogen, ant-mpt).

### Vector constructs
The CRISPR–Cas9 plasmids that target *HLF* were constructed based on the lentiCRISPR v2 (Addgene, 52961) while the ones targeting *HLF* and *LPXN* enhancer region were constructed based on the pLV-U6-gRNA-UbC-eGFP-P2A-Bsr (Addgene, 83925). The shRNA plasmids that target *HLF* / *SMARCA4* were acquired from sigma, while the shRNA plasmids that target *LPXN* / *AJAP1* were constructed based on the pLKO.1 puro backbone (Addgene, 8453). pLX304-*B4GALT6*/ *TRMT5*/ *CYCS*/ *CD44*/ *ICAM2*/ *LPXN*/ *PVRL1* (*NECTIN1*)/ *ADAMTSL1* were requested from UNC, while pLX304-*HLF* (EX-G0077-LX304-B) were bought from GeneCopoeia. For pLX304-*FOSB*/ *AJAP1*/ *HTRA1*, pDONR201-*FOSB* (HsCD00001737)/ pDONR221- *AJAP1*(HsCD00877134)/ pDONR221-*HTRA1*(HsCD00860462) were acquired from DNASU, and were cloned into pLX304 backbone (Addgene, 25890). For pInducer20-*LPXN*, pDONR221-*LPXN* (HsCD00041693) was acquired from DNASU, and was cloned into pInducer20 backbone (Addgene, 44012). For pInducer20-PXN-Y118E/Y118F mutants, pLV-Paxillin was acquired from Addgene (#176106), and was mutated and cloned into pInducer20 backbone. *LPXN* promoter reporter plasmid was bought from GeneCopoeia (HPRM35884-LvPF02), while pLVX-*SMARCA4* plasmid was acquired from Rugang Zhang at the Wistar Institute in Philadelphia. All related primers for plasmid construction are shown in Supplementary Data 2.

### Virus Infection
For lentiviral plasmids, 293 T cells were used for virus production by co-transfection of the psPAX2 (Addgene, 12260) and pMD2.G (Addgene, 12259) in Opti-MEM with the addition of Lipofectamine 3000. 0.45 μm filters were then applied during the virus collection step. Cells were seeded in 6-well plates for 30-50% confluency, and 200 μl virus per well was added while 6-8 μg/mL polybrene was used to increase infection efficiency. Media were replaced after 12–24 h infection, and cells were cultured for another 1 - 2 days followed by an appropriate antibiotic selection (Puromycin 2 μg/mL for sgRNA/shRNA plasmids, Blasticidin 10 μg/mL for pLX-304 backbone plasmids, G418 800 μg/mL for pInducer-20 backbone plasmids).

### CRISPR screening
**Preparation of library virus.** Human GeCKOv2 CRISPR knockout pooled lentiviral library (Addgene, 1000000048) was employed for the Genome-wide CRISPR screen of essential genes regulating tumor

metastasis in vivo. The library was amplified on large LB agar plates followed by plasmid purification. Library PCR was performed to amplify and barcode the sgRNA sequences with Illumina adaptors followed by NGS sequencing to check sgRNA distribution. To produce the library virus for infection, pooled lentiviral vectors and packaging system as shown above were applied for virus production.

**CRISPR screen in vivo.** $1 \times 10^8$ UMRC2 or A498 ccRCC cells were infected with the library viruses at a multiplicity of infection (MOI) of ~0.3 and selected with 2 μg/mL puromycin for 72 h. After confirmation of optimal tittering, the remaining cells were expanded for a week with 1 μg/mL puromycin treatment, followed by subcutaneous injection into each flank of NSG mice (Jackson lab, female, 6 ~ 8 weeks of age) at $1.5 \times 10^7$ cells per flank. With this approach, a minimum of 200x library coverage per mouse was achieved, with >200 injected cells incorporating an identical single guide RNA (sgRNA) of the GeCKOv2 library. A total of 5 mice with similar library coverage were used as biological replicates for each ccRCC cell line. After eight weeks, mice were sacrificed, and xenograft tumors and lungs were collected for further analysis.

**Bulk extraction of genomic DNA.** To extract genomic DNA from harvested tissues, a previously verified protocol[55] was followed with minor modifications. Briefly, all samples were individually homogenized by using a ceramic mortar and pestle on dry ice. 200–500 mg of tissue powers were lysed at 55 °C overnight in lysis buffer (50 mM Tris, 50 mM EDTA, 1% SDS, pH=8.0) supplemented with Proteinase K. Tissue lysates were then incubated with RNase A to digest RNA, and with 2.5 M ice-chilled ammonium acetate for protein precipitation, followed by isopropanol precipitation. PCR was performed to amplify and barcode the genomic DNA as shown above, and the final purified products were sequenced on a HiSeq 2500 (Illumina).

**Identification of genes involved in lung metastasis.** Raw reads were first mapped to a reference list of all GeCKOv2 sgRNA sequences and then quantified in each tissue sample. MAGeCK v0.5.4[56] was applied to process the sgRNA read counts, generating fold change (FC) and *p*-values for enrichment analysis of lung metastatic lesions relative to corresponding primary tumors. To incorporate data from multiple biological replicates, we performed the following steps: 1) We utilized the paired test function of MAGECK by comparing the sgRNA enrichment of metastatic tumors with primary tumors in the same mouse; 2) We chose to compare *p*-values instead of FC, because FC was prone to show relatively high variations; 3) Out of 5 mice, we only incorporated the data from 4 mice with higher preferences for lung metastasis (because ccRCC metastasize to the lung (70%) much more frequently than to the liver (18%) and other sites[2], therefore genes consistently regulate lung metastasis may be more pathologically relevant); 4. Out of 5 mice, the smallest adjusted *p*-values were chosen as the representative score (essentiality scores) and subjected to the final ranking analysis. The top ten genes with highest essentiality scores for both UMRC2 and A498 cell lines were selected for further examination.

In addition, we also performed an alternative analysis by incorporating negative *p*-values from all biological replicates, after application of the following data filtering criteria: the negative rank ("neg| rank" within the MAGeCK report) for lung metastases is higher than that of pairing liver metastases, and the positive rank ("pos|rank" within the MAGeCK report) for lung metastases is simultaneously lower than that of pairing liver metastases. These criteria were included to identify genes consistently regulating lung but not liver metastasis in all biological replicates, given that renal tumors metastasize to the lung much more frequently than to the liver[2]. Based on this alternative analysis, *HLF* was comprehensively ranked at #3 out of all screened genes in UMARC2 and A498 renal cancer cell lines.

## Quantitative real-time PCR

For reverse transcription quantitative PCR (RT-qPCR), total RNA was extracted using the RNeasy mini kit, and cDNA synthesis was performed using the iScript™ cDNA Synthesis Kit. RT-qPCR was conducted using the CFX384 Real-Time PCR System (Bio-Rad). Relative amplicon expression was calculated by applying the $2^{-\Delta\Delta Ct}$ method. The sequence of RT-qPCR primers can be found in Supplementary Data 2. For ChIP-PCR, the DNA samples acquired from ChIP experiment were used as the templates, and the primers used were designed based on the binding sequence from sequencing data. The sequence of ChIP-PCR primers can be found in Supplementary Data 2. The data were analyzed by using the following formula to indicate the abundance of protein enrichment: percent input = $100\% \times 2^{(CT\ input\ sample\ -\ CT\ IP\ sample)}$.

## Protein extraction and western blotting

For common protein preparation, BEC buffer (0.1 mM EDTA, 120 mM Nacl, 50 mM Tris −HCL pH8.0, 0.5% NP-40 and 10% Glycerol) supplemented with phosphoSTOP tablets (Roche) and protease inhibitor cocktail (Roche) was used for cell lysis. However, cell lysates used to detect endogenous HLF by applying generated antibodies were treated with extra 1% SDS and 25 U/mL Benzonase® Nuclease (sigma, 70664-3) for 10 mins rotation at RT. For tissue samples, TissueRuptor II (QIAGEN) was used for homogenization. The concentration of protein was measured by Bradford assay and its amount was equalized in 5x loading buffer followed by heating. Proteins were loaded and resolved with SDS-PAGE, followed by membrane transferring. Normally after blocking with 5% non-fat milk in TBST (Tris-Buffered Saline with 0.1% Tween 20), the NC membranes were incubated with primary antibodies overnight and secondary antibodies the next day. Finally, chemiluminescent substrates ECL or SuperSignal™ West Femto maximum sensitivity substrate were used to acquire protein bands by applying the ChemiDoc Imaging System (Bio-Rad) and analyzed using the Image Lab software. Antibodies used are as below: V5-Tag (1:1000 dilution, Cell Signaling Technology, 13202S), α-Tubulin (1:1000 dilution, Cell Signaling Technology, 3873S), Vinculin (1:1000 dilution, Sigma-Aldrich, V9131), HA-Tag (1:1000 dilution, Cell Signaling Technology, 3724S), Flag-Tag (1:1000 dilution, Cell Signaling Technology, 14793S), H3 (1:1000 dilution, abcam, ab1791), Dnmt1 (1:200 dilution, Santa cruz, sc-271729), H3K27ac (1:1000 dilution, Cell Signaling Technology, 4353S), H3K9ac (1:1000 dilution, abcam, ab10812), H3K4me3 (1:1000 dilution, abcam, ab8580), LPXN (1:1000 dilution, Lsbio, LS-C313296-100), SMARCA4/Brg1 (1:1000 dilution, Cell Signaling Technology, 49360S), Paxillin (1:1000 dilution, Cell Signaling Technology, 2542S), Phospho-Paxillin (Tyr118) (1:1000 dilution, Cell Signaling Technology, 69363S), and HIF-1β/ARNT (1:1000 dilution, Cell Signaling Technology, 3414S). HRP-conjugated goat anti-mouse (1:5000 dilution, Thermo Fisher Scientific, 31430) and HRP-conjugated goat anti-rabbit (1:5000 dilution, Thermo Fisher Scientific, 31460) were used as secondary antibodies. The specificity and quality of all commercial antibodies have been confirmed. HLF antibodies (N/C terminal) are designed and generated by Lu Wang from Northwestern University, and the antibodies have been firstly tested in HepG2 cells with *HLF* deletion by sgRNA. Quantification of protein bands was performed by using Image J software.

## RNAscope chromogenic in situ hybridization assay

RNAscope assay (RNAscope 2.5 LSx Reagent Kit − RED, Cat# 322750) was performed according to the protocol from Advanced Cell Diagnostics (ACD) and conducted in the UTSW Tissue Management Shared Resource Core Facility using the Leica Biosystems BOND RX automated IHC/ISH slide staining systems (Leica Software v BDZ 15 or BXD15). The experimental condition was optimized as: RNAscope 2.5 LSx RED ISH (default Amp5 15 mins) and RNAscope 2.5LSx Target Retrieval (95) (default 15 min). To validate the specificity of the HLF probe, paired normal and ccRCC tumor tissues were utilized. These

tissues had previously been confirmed to express lower *HLF* mRNA levels in the tumor tissues through RT-qPCR analysis. Homo sapiens peptidylprolyl isomerase B (cyclophilin B) (PPIB, Cat# 313908) and complete CDS of dihydrodipicolinate reductase (dapB, Cat# 312038) gene were used as positive control probe and negative control probe respectively. Images were taken using the widefield Nikon Ti for epifluorescence at 100x. Quantification was performed based on the methodology of heterogeneous target expression according to the guideline from ACD, and quantification was calculated as below: H-Score = 0*(% of cells in bin 0) + 1*(% of cells in bin 1) + 2*(% of cells in bin 2) + 3*(% of cells in bin 3) + 4*(% of cells in bin 4), bin 0 (0 dots/cell), bin 1 (1–3 dots/cell), bin 2 (4–9 dots /cell), bin 3 (10–15 dots/cell), bin4 (>15 dots/cell).

## Immunohistochemistry (IHC) and Hematoxylin and eosin (H&E) staining

Both IHC and H&E staining were conducted by the Tissue Management Shared Resource Core at UT Southwestern Medical Center. For IHC, a custom rabbit polyclonal antibody against HLF was used for IHC staining. Briefly, after dewaxing and hydrating, antigen retrieval was performed for 20 min at 100 °C in pH9 EDTA buffer, slides were incubated with HLF (1:3000) primary antibody for 20 min, followed by secondary antibody incubation (1:1000). H-score analysis was performed by pathologist Liwei Jia from the department of Pathology at UT Southwestern Medical Center. Images (20x) were taken by using the Nikon upright microscope at the department of Pathology at UT Southwestern Medical Center.

## Flag tag knock-in

gBlocks of 500 ~ 600 bp homology arms were synthesized in IDT corporation, followed by PCR amplification. The pFETCh_Donor (EMM0021, Addgene #63934) backbone vector was digested using BsaI and BbsI, followed by single step gibson assembly reaction with amplified gBlocks. Sanger sequences from both ends of each homology arm were applied for verification. The CRISPR–Cas9 gRNA plasmid was constructed based on the pSpCas9n(BB)–2A-Puro (PX462) V2.0 (Addgene #62987). Generating Flag-KI stable cell line, a ratio of 2:1 of pFETCh_Donor and Cas9 gRNA was applied for simultaneous transfection, followed by G418 selection and single colony selection.

## LPXN promoter reporter assay

LPXN promoter reporter plasmid was bought from GeneCopoeia (Cat# HPRM35884-LvPF02), specifically including the HLF binding sequence identified through HA-tagged HLF ChIP-seq analysis. 293 T cells were used for packaging virus, 786-O pLX304-empty vector and pLX304-HLF stable cells were infected with the virus separately, followed by puromycin selection. Stable reporter cells were then seeded into MatTek 35 mm glass bottom dish (NC9341562). After 1 day, images were taken by applying the Zeiss LSM 780 Confocal Microscope, and 20x images were used for quantification while 40x/1.3 Oil images were used for presentation.

## 3D collagen invasion assay

Indicated cell lines (1000 cells per well) were seeded in the Nunclon™ Sphera™ 96-Well plate (Thermo Scientific, 174925) and cultured for 7 days to generate spheroids. For the invasion assay, the spheroids were pelleted (200 g, 2 min) and embedded in 3D collagen I in the 4-well μ-slide (Ibidi, 80427), according to the instructions of the 3D Collagen Culture Kit (sigma, ECM675). Images of spheroids were taken using the microscope (Echo) during indicated days (2 ~ 7 days) based on cell types. Quantification was done using the ImageJ software (Fiji). For confocal imaging, 100 cells (50 cells of RFP-labeled 786-O sgCtrl, 50 cells of GFP-labeled 786-O sg*HLF*) per well were seeded in the Nunclon™ Sphera™ 96-Well plate to form spheroids, followed by embedding in 3D collagen I in the 4-well μ-slides. The spheroids were imaged using the Z-stack imaging of the Zeiss LSM 780 Confocal Microscope.

## Single cell migration tracking

Stiff collagen substrates were generated by coating the glass region of the 35 mm dish (P35G-1.5-14-C, MatTek) or glass-bottom 96-well black plate (Cellvis, P96-1.5H-N) with bovine collagen type I (PureCol EZ Gel, 5 mg/ml, Advanced Biomatrix). After coating with diluted collagen solution (50 μg/ml in PBS), the dishes/plates were incubated at 37 °C for 1 h before seeding cells. The dishes/plates were also coated with 0.10% (w/v) Poly-L-lysine (P8920, sigma) following a 5 min incubation and drying the surface at room temperature before seeding cells. After seeding cells ($1 \times 10^4$ cells in 2.5 mL in 35 mm dish and 500 cells in 200 μL in 96-well plate), the dishes/plates were incubated overnight for stable cell attachment. Live cell imaging was performed using either a DeltaVision Ultra microscope system (GE Healthcare) or ImageXpress Confocal HT.ai High-Content Imaging System (Molecular Devices) at 37 °C supplied with 5% CO2. Images were captured at 10 min intervals for 16 h using a 20x objective (DeltaVision) or 10x objective (ImageXpress) with $7 \times 1.5$ μm z-sections. For data analysis, the ImageJ Manual Tracking Plugin was used to track cell motility and tab-delimited text files were imported into the Chemotaxis and Migration Tool 2.0 (Ibidi) for plotting and quantitative analysis of Euclidean distance, velocity, and directionality. Statistical analyses were performed using GraphPad Prism 9.0 software.

## Immunofluorescence staining

F-actin staining was performed as described previously[35]. Briefly, cells were seeded on glass bottom 4-well μ-slide (Ibidi, 80427) coated with 0.10% (w/v) Poly-L-lysine or 50 μg/ml collagen I ($1 \times 10^4$ cells / 700 ul / well). After cell culturing, cells were rinsed with PBS, fixed with 4% paraformaldehyde (PFA) for 10 min, washed with PBS, permeabilized with 0.5% Triton-X100 for 10 min, washed with PBS, and blocked in 3% non-fat dry milk in PBS at RT for 1 h. The cells were then washed with PBS and stained with 1% BSA-diluted Alexa Fluor™ 568 Phalloidin (A12380, Thermo Fisher Scientific) for 20 min at RT. Cells were rewashed, followed by DAPI incubation for 5 min and another wash. For staining with phospho-Paxillin (Tyr118), p-PXN-118 (Cell Signaling Technology, 69363S) and LPXN (ATLAS ANTIBODIES, HPA061441), permeabilized cells were incubated with 5% BSA in PBS for 30 min at RT, followed by incubation with p-PXN-118 (1:400 dilution) or LPXN (1:500 dilution) overnight at 4 °C. The next day, cells were washed, incubated with secondary antibody (Donkey anti-Rabbit IgG ReadyProbes Alexa Fluor™ 488, Invitrogen, R37118) at 1:1000 dilution in PBS for 1 h at 37 °C in dark. Cells were washed and continued with Phalloidin and DAPI staining as above. F-actin and p-PXN-118 were visualized using a Revolve Fluorescence Microscope (Echo) at 40× objective. The average intensity was quantified using the ImageJ software (Fiji), and the average filament number and F-actin length were quantified by applying the FilamentSensor 2.0 based on Java[57]. For GFP and RFP staining in PDX tumor tissues, freshly dissected tumors were rinsed with cold PBS, embedded in OCT, and quickly frozen using dry ice, followed by cryosectioning at 30 μm. Using a hydrophobic pen, slides were fixed in 4% PFA for 10 min at room temperature, then the fixative was aspirated, and samples were rinsed twice with PBS. Samples were then permeabilized with 0.5% Triton X-100 for 10 min, followed by aspiration and two PBS rinses. Next, samples were incubated with 5% BSA in TBST for 30 min at room temperature, rinsed twice with PBS, and then incubated with primary antibodies (anti-RFP, proteintech, 67378-1-Ig; anti-GFP, proteintech, 50430-2-AP) at a 1:500 dilution in 5% BSA and 0.1% Triton X-100 for 24 h at 4 °C in the dark. After rinsing with PBS, samples were incubated with fluorochrome-conjugated secondary antibodies (Donkey anti-Rabbit Alexa Fluor™ 488, Invitrogen, R37118; Goat anti-Mouse Alexa Fluor™ 647, Invitrogen,

A21235) at a 1:1000 dilution in PBS for 1 h at 37 °C in the dark. Following two PBS rinses, samples were incubated with DAPI for 5 min, briefly rinsed, mounted, and then analyzed via confocal imaging.

## Enhancer activation and inhibition

To generate the enCRISPRa and enCRISPRi stable cell lines, the cells were transduced with lentivirus expressing pLV-dCas9-p300-P2A-PuroR (Addgene #83925) and pLV-dCas9-KRAB-PGK-HygR (Addgene #83890) respectively, following a treatment with corresponding selection marker. For enCRISPRi or enCRISPRa experiments, the cells were transduced with sgRNA lentiviruses targeting specific enhancer sequence or nontargeting sequence, followed by Blasticidin selection, and subsequent qRT-PCR analyses.

## Cell competition assay

786-O control cells (WT) and HLF knock-out cells (HLF KO) were infected with lentivirus expressing RFP (Plenti6 mRFP) and GFP (Pll7.0 mU6-Ubc-mEGFP), respectively. $1 \times 10^6$ WT cells and $1 \times 10^6$ HLF KO cells were seeded together in 6-well plates and imaged every other day using a Revolve fluorescence microscope (Echo). % area per channel was quantified using the ImageJ software (Fiji).

## Cell proliferation assay/Cell viability assay

For MTS assay, 1000 cells per well were seeded in 96-well plates in appropriate medium. MTS reagents (Promega) supplemented in medium (1:10 ratio) were treated with cells at indicated time points, incubated at 37 °C for 1 h, followed by OD absorbance measurement using the microplate reader (BioTek). For cell viability assay, 5000 cells per well were plated in 96-well plates in appropriate medium. After overnight incubation, a serial dilution of AU-15330 was prepared and added to the plate. The cells were then incubated for 72 h, after which MTS reagent was applied as described to assess cell proliferation. The EdU assay was performed according to the Click-iT™ EdU Cell Proliferation Kit (Invitrogen, C10337) instructions. Briefly, cells were seeded and cultured overnight on a 35 mm glass bottom dish, incubated with 10 μM EdU for 8 h, and fixed in 4% paraformaldehyde for 15 min. The cells were washed with 3% BSA, followed by incubation with 0.5% Triton X-100 for 20 min, rewashed and applied Click-iT® reaction cocktail for 30 min, washed again and stained with Hoechst 33342. Images were taken using a Revolve Fluorescence Microscope (Echo) at 20× objective.

## 2D and 3D colony formation assay

For 2D colony formation assay, 2000 cells per well were seeded in 6-well plates in appropriate medium. Cells were stained with 0.5% crystal violet for 1 h at an appropriate density. For 3D colony formation assay, ~20,000 cells were seeded in the upper layer agar, cultured for 4 weeks, and stained with 100 μg/mL iodonitrotetrazoliuim chloride solution. The quantification of colonies was performed using ImageJ software.

## RNA-seq and analyses

Total RNA was extracted by using the RNeasy® Mini Kit (Qiagen, 74106) according to corresponding handbook, and RNase-Free DNase Set (Qiagen, 79254) was applied for on-column digestion of DNA during RNA purification. The library preparation, quality control and sequencing were performed in the Novogene Corporation. The data was analyzed as described previously[58]. Raw data has been deposited in the Gene Expression Omnibus (GEO) under accession number GSE272563. GSEA analysis and GO analysis were performed by using WebGestalt (WEB-based Gene SeT AnaLysis Toolkit). The clinical RNA-seq data of normal and ccRCC tumor was obtained from Dr. Payal Kapur in the Pathology department at UT Southwestern Medical Center. The comparison analysis and graph plotting were performed using Prism 9.0.

## ChIP-seq and analyses

ChIP (Chromatin immunoprecipitation) was performed on 786-O cells with anti-FLAG (Cell Signaling Technology, 14793) and anti-HA (Cell Signaling Technology, 3724) antibody as previously described[27]. Briefly, the samples were cross-linked with 1% formaldehyde, followed by 125 mM glycine treatment. Samples were then lysed in lysis buffer (10 mM Tris-HCl pH 8.0, 100 mM NaCl, 1 mM EDTA, 0.5 mM EGTA, 1% Triton x-100) and sonicated with BioRuptor. Dynabeads were incubated with FLAG or HA antibody at 4 °C overnight with rotation followed by overnight incubation with sonicated samples. The samples were then washed and collected by applying a magnetic stand followed by elution and reverse crosslink. RNase A (sigma, 70856-3) was used for RNA degradation, and Proteinase K (sigma, 3115828001) was used for protein degradation. DNA was purified using the Qiagen PCR purification kit (Qiagen, 28106). DNA libraries were prepared using the NEBNext® Ultra™ II DNA Library Prep Kit (NEB, E7645S) followed by followed by sequencing in the CRI's Sequencing Facility at UT Southwestern. The HA-HLF ChIP-seq data has been deposited in the GEO under accession numbers GSE272563. The ChIP-seq data of H3K4me1 and H3K27ac in parental 786-O and 786-O M1A cells were obtained from GEO under the accession code GSE98015[59].

## ATAC-Seq

Sample preparation for the assay of transposase-accessible chromatin with sequencing (ATAC-Seq) was performed according to the Manual of the ATAC-Seq Kit (Active Motif, 53150). Briefly, cells were collected, pelleted, and resuspended in ATAC lysis buffer. Cells were pelleted again, followed by tagmentation reaction in the master mix and purification by applying DNA purification binding buffer and column. Tagmented DNA was amplified by PCR using the i5/i7 indexed primer, followed by sequencing at the Next Generation Sequencing core facility in UT Southwestern. The ATAC-seq data has been deposited in the GEO under accession number GSE272563.

## Animal experiments

**Orthotopic transplantation.** Luciferase-labeled cell lines were used for kidney orthotopic injection. Six-week-old NSG mice (Jackson Lab) with an equal ratio of males and females were used. Briefly, $1 \times 10^6$ 786-O cells transduced with indicated plasmids or $5 \times 10^5$ PDX cells transduced with indicated plasmids were resuspended in 20 μL Matrigel (1:1 dilution), followed by orthotopic injection into the left kidney of each mouse. SPECTRAL AMI-HTX imaging system was applied weekly to monitor tumor growth by intraperitoneal injection (IP) of luciferin (15 mg/mL). For inducible pInducer20-HLF stable cell line, mice were fed Purina rodent chow doxycycline at week-3 (The luciferin signal exhibited a stable increase). At the final time point, mice were sacrificed, and blood was collected to calculate the circulating tumor cells (CTCs) by measuring the luciferin signal (applies only to the EV/pLX304-HLF group). Tumors were extracted to be weighed, and lungs were collected to perform ex vivo imaging in 24-well plate using 300 μg/mL luciferin solution.

**Subcutaneous transplantation.** Six-week-old NSG mice (Jackson Lab) with an equal ratio of males and females were used for subcutaneous transplantation. $2 \times 10^6$ luciferase-labeled cells suspended in 100 μL PBS were injected directly into the back of skin flaps. The size of the tumors was measured by using a digital electronic caliper every 5 days, and tumor volume was calculated based on the formula: $V = 0.5 \times L \times W^2$, where V is the tumor volume, L is the tumor length, and W is the tumor width. At the final time point, mice were sacrificed, and tumors were collected to be weighed while lungs were collected to perform ex vivo imaging described as above. The maximal tumor size permitted by the ethics committee was no more than 2 cm for single tumor or cumulative diameter of 3 cm for multiple tumors, and the maximal tumor size did not exceed the limit.

**Intravenous injection.** Luciferase-labeled cell lines were used on six-week-old female NSG mice (Jackson lab) for intravenous tail vein Injections with $1 \times 10^6$ cells suspended in 100 μL DMEM medium. SPECTRAL AMI-HTX imaging was performed immediately after the injection, and lung signals were monitored by weekly imaging afterwards. At the final time point, mice were sacrificed, and lungs were collected to perform ex vivo imaging described above.

All mice were housed in the ARC Mouse Facility at UT Southwestern under specific-pathogen-free (SPF) conditions, with climate control and a 12 hour light/dark cycle.

### Isolation of orthotopic tumor cells
The mice with orthotopic injection of 786-O cells were sacrificed at week-8. Kidney primary tumors and lung mets tissue were collected, washed with cold PBS and chopped with sterile razor blade/scissors, followed by incubation with 1 mg/ml collagenase (type 4) in serum-free DMEM media for 1 - 2 h at 37 °C with shaking. Tissues were pelleted (200 g, 3 min), washed with PBS, resuspended with trypsin EDTA for 5 min at 37 °C, pelleted and rewashed, filtered through the 100 μm EZFlow cell strainer, pelleted and resuspended in DMEM medium supplemented with 10% FBS and 2% pen/strep, and cultured in the incubator for several passages to yield purified 786-O cells. The cells isolated from primary tumor were indicated as 786-O-P, while the cells isolated from the lung were 786-O-LM.

### Statistics and reproducibility
No statistical methods were used to predetermine sample size, but at least three samples were used per experimental group. Samples and experimental mice were randomly assigned to experimental groups. An unpaired two-tailed Student's t-test was used to compare two groups, while one-way ANOVA followed by (if significant) a post hoc Dunnett-t-test was performed for multiple groups. All statistical analyses were performed using Prism 9.0 (GraphPad Software). The data for graph plotting were presented as the mean±SEM, while individual samples were represented as dots in certain graphs. Precise $P$ values are shown in the figures, and $P$ values less than 0.05 were considered statistically significant. All experiments presented in the manuscript were repeated in at least two independent experiments or biological replicates with similar results.

### Reporting summary
Further information on research design is available in the Nature Portfolio Reporting Summary linked to this article.

## Data availability
The raw data of CRISPR screening generated in this study has been deposited in NCBI under accession number PRJNA1161221. The ChIP-seq, ATAC-seq and RNA-seq data generated in this study have been deposited in the Gene Expression Omnibus (GEO) database under accession code GSE272563. The ChIP-seq data of H3K4me1 and H3K27ac in parental 786-O and 786-O M1A cells were obtained from GEO under the accession code GSE98015. The clinical RNA-seq data of normal and ccRCC tumors was deposited in European Genome-phenome Archive (EGA) database under the accession number EGAS00001005516. The processed RNA-seq data are provided in Supplementary Data 1. All the other data supporting the findings of this study are available within the article, Supplementary information file, or source data file. Source data are provided with this paper.

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

## Acknowledgements

This work is supported by National Science Foundation of China (B. Li, 82192892 and 82472846), Cancer Prevention and Research Institute of Texas (CPRIT) (Q. Zhang, RR190058), National Cancer Institute (Q. Zhang, R01CA294133, M. C. Simon, 2R35CA220483, J. Brugarolas, P50CA196516), AACR-Exelixis Renal Cell Carcinoma Research Fellowship (J. Zhou, 22-40-66-ZHOU), Department of Defense Kidney Cancer Research Program (M. C. Simon, HT94252310859), National Institute of General Medical Sciences (L. Wang, R35GM146979). We acknowledge the assistance of the University of Texas Southwestern Tissue Management Shared Resource, a shared resource at the Simmons Comprehensive Cancer Center, supported in part by the National Cancer Institute under award number P30 CA142543. We acknowledge the assistance of the Preclinical Radiation Core Facility supported by CPRIT (RP180770). We thank Dr. Yanan Wang for providing EGFP plasmid. We thank Dr. Tao Yue for providing 143B cell line, and kidney cancer patients for donating their tissues for this study.

## Author contributions

J.Z., M.C.S., B.L. and Q.Z. designed experiments. J.Z. performed experiments. A.H. and J.M.S. performed computational analyses. K.K., H.Z., P.K. L.X., L.D. and S.M. helped with clinical patient data analysis. Q.H., L.H., C.Z., C.L., Y.A., H.F., T.W., Q.L., H.L., M.T., J. F. and F. Z. helped with in vitro experiments. H.Y. and CH.Z. helped with animal experiments. L.W. helped with custom antibody generation. J.Z., A.F., A.H., J.M.S., Q.H., P.L. and J.B. wrote and revised the manuscript.

## Competing interests

Dr. Qing Zhang received the consultation fee from Exelixis. Other authors declare no competing interests.
