## [Transparent Peer Review file · Nature Communications]

SWI/SNF ATPase silenced HLF potentiates lung metastasis in solid cancers

Corresponding Author: Professor Qing Zhang

Version 0:

Reviewer comments:

Reviewer #1

(Remarks to the Author)

Reviewer comments for NCOMMS-24-72086-T

“SWI/SNF ATPase - silenced HLF potentiates lung metastasis in solid cancers”

In this manuscript the authors use subcutaneous injection of clear cell renal cell carcinoma (ccRCC) cells and a genome wide CRISPR screen to identify a novel suppressor of metastasis – Hepatic leukemia factor or HLF. Of note the subcutaneous injections used for the genome wide CRISPR screen are less than ideal for spontaneous metastasis studies; however, the authors perform several validation experiments using orthotopic injection of ccRCC cells. The authors also continue by providing substantial mechanistic detail describing how HLF influences cell invasion by modifying the way in which cells integrate signals from the collagen extracellular matrix (ECM). HLF regulates levels of LPXN to inhibit cell adhesion signaling through paxillin and the actin cytoskeleton dynamics necessary for cell invasion. In addition, the authors determine how HLF is downregulated in metastatic cells by looking at HLF enhancer regions to discover that SMACA4 is a major repressor of HLF transcription. These findings suggest that inhibition of SMARCA4 activity might be effective at limiting metastatic disease in ccRCC.

Overall, this is an extensive and comprehensive body of work that is experimentally well designed with robust data of high quality. Multiple cell lines, in vitro and in vivo models are used to support the manuscript's main conclusions. Moreover, various methodologies are used to investigate molecular mechanisms including gain and loss of function studies as well as next generation sequencing based methods to explore genome-wide and HLF-specific regulation of gene expression (RNA-seq, ATAC-seq, ChIP-seq). Finally, the work is somewhat novel, and simultaneously holds great potential to provide broad clinical benefit for patients with various types of cancer using a SMARCA4 inhibitor (PROTAC-targeted degradation of SMARCA4) that has shown efficacy in targeting prostate cancer cells and that is currently in phase I clinical trials for metastatic uveal melanoma and advanced hematologic malignancies. For the above reasons, I recommend this article as suitable for publication in Nature Communications. However, I also include the following minor comments for consideration of their potential to add clarity and refinement to the current reported data.

Minor comments:

1. Previous reports cited in this manuscript (ref 12 and 13) indicate that HLF can activate c-JUN and inhibit p65-NFkappaB activity in hepatocellular carcinoma and non-small cell lung cancer respectively. Is there evidence for similar regulation of these factors in the ccRCC models and if yes, could their potential impact on metastasis be discussed?

2. Continuing from the comment above, recent reports suggest that HLF can activate YAP1 activity in ovarian cancer to promote cancer cell stemness, and induce levels of Tensin 1 in prostate cancer to modulate prostate cancer cell survival, invasion and metastasis. It would be informative to comment on these studies and any potential relevance to the current mechanism described for ccRCC, which involves adhesion mediated integration of mechanical cues from the ECM to the actin cytoskeleton.

Han T, Chen T, Chen L, Li K, Xiang D, Dou L, Li H, Gu Y. HLF promotes ovarian cancer progression and chemoresistance via regulating Hippo signaling pathway. *Cell Death Dis.* 2023 Sep 14;14(9):606. doi: 10.1038/s41419-023-06076-5. PMID:

Hao Zhou, Fang Wang, Tensin 1 regulated by hepatic leukemia factor represses the progression of prostate cancer, *Mutagenesis*, Volume 38, Issue 6, November 2023, Pages 295–304, <https://doi.org/10.1093/mutage/gead027>

3. In some instances, images of the lung bioluminescence appear “cut off” and fail to show sufficient coverage of the mouse ventral midsection. Is there an explanation for this? Alternatively, it would be advisable to include some IHC (e.g. H&E) of lung metastases in addition to the bioluminescence imaging (in vivo and ex vivo) to support the robust quality of these data. Lung sections could also be used to further validate the proposed mechanism where appropriate.
4. While current data support a role for HLF in modulating collagen derived mechanical cues in driving ccRCC cell invasion (3D collagen and “2D stiff collagen” models), these data would be better supported by the culture of cells on a range of stiffnesses that correspond to normal renal ECM environments as well as soft and stiff disease states. For example, these studies could be accomplished using polyacrylamide hydrogels prepared with different crosslinker concentrations/stiffnesses followed by conjugation with collagen ECM. Such studies would also permit evaluation of the HLF-LPXN-Paxillin axis in different mechanical environments.
5. It would be nice to see evidence for differential LPXN localization at adhesion sites in cells with high vs. low levels of HLF. This would strengthen the mechanism of HLF induction of LPXN to limit Paxillin phosphorylation and adhesion signaling.
6. It is unclear whether the sample population size (n) is reported for TNMplot data, for example in Figure 1h.
7. In Figure 4 and Extended Data Figure 5, it appears that ICAM2 and ADAMTSL1 have similar impacts on cell invasion as compared to AJAP1. An additional rationale for LPXN selection and elimination of these targets from subsequent analyses would be helpful.
8. Extended Data Figure 6f appears to be missing protein labels for the left panel of immunoblots.

Reviewer #2

(Remarks to the Author)

In this manuscript, Zhou and colleagues perform an in vivo genome-wide CRISPR-Cas9 screen in ccRCC to identify factors that govern lung metastasis. They identify HLF as a novel tumor suppressor, whose silencing in primary tumor promotes invasion and metastasis. The authors performed a detailed and elegant molecular dissection of the mechanism of action of the HLF-LPXN-PXN axis in regulating cell-matrix adhesion and metastasis. Collectively, the authors have conducted a very thorough analysis of the role of HLF in regulating metastasis of ccRCC. I hope that the following comments – mostly regarding the screen - will help authors improve their manuscript.

Comments and technical questions regarding the in vivo screen:

1. Setting: the authors perform a screen for lung metastasis by injecting two human ccRCC lines harboring a genome wide KO library subcutaneously into NSG mice. This setting is not ideal for a screen that aims at identifying factors that regulate metastasis because the molecular requirement for invasion and metastatic dissemination from the skin might differ from those from the kidney. Moreover, the use of NSG mice prevents analysis of the immune regulation of the metastatic cascade. While this might be advantageous to focus on non-immune effects, it is still a reductionistic view of metastasis. The authors should comment on why they chose the subcutaneous setting for the screening rather than the orthotopic model used for validation.
2. Coverage at transduction: authors state that they transduced 1×10^8 cells with the library (1×10^5 sgRNAs). This would mean a 1000x coverage if 100% of the cells were transduced, which is not the case with a MOI of 3%. What was the transduction efficiency and therefore the actual coverage at the time of library transduction?
3. Coverage at engraftment: authors state that they injected 1.5×10^7 cells per flank for a coverage of 200x. However, $1.5 \times 10^7 / 1 \times 10^5$ is 150x. Did the authors inject mice in both flanks?
4. Both sgRNAs and Cas9 are expressed constitutively in cell lines upon transduction. Therefore, during the subsequent 1-week culture in vitro of the cells, sgRNA distribution will be altered compared to the viral library (e.g. sgRNAs targeting essential genes will be lost). The abundance of sgRNAs targeting genes that regulate expansion in culture (proliferation) will also be altered. Did authors sequence the library in cells prior to injection? This would help in normalizing the results for these effects (by setting a baseline distribution) and would determine the sgRNA distribution and coverage of the library that was ultimately screened in vivo. Moreover, authors could compute depleted sgRNAs in metastases vs pre-injection and depleted sgRNAs in primary vs pre-injection to identify genes that are specifically needed for metastasis but not for primary tumor growth (see next point). In any case, authors should show depletion of sgRNAs targeting essential genes and distribution of control sgRNAs.

5. In addition to altering library distribution in vitro, the effect of the induced perturbation is going to affect every subsequent step of the screen in vivo: from engraftment to tumor growth, to dissemination, survival in circulation, colonization and growth at secondary site. All these processes are being screened at the same time. Engraftment by itself causes a massive bottlenecking effect. Perturbations that impact engraftment and tumor growth will be depleted or enriched during the first weeks of tumor growth in vivo, prior to metastatic dissemination. After colonization, perturbation will be also impacting growth at secondary site. While this can only be avoided using an inducible system, some controls and additional analyses can be conducted to estimate the bottlenecking and skewing of the library. Did the authors sequence the sgRNA library in tumors also upon the engraftment period of this model but prior to metastatic dissemination? This would be the ideal baseline for comparison. Can the authors identify which sgRNAs are lost between plasmid or viral library and experimental endpoint?

6. Authors compare sgRNAs depleted in metastases vs primary tumors and identify sgRNAs enriched in metastasis whose inferring genes are metastasis suppressors. Did the authors also find sgRNAs depleted in metastasis but not in primary tumors that would hint at genes required for metastasis? Were sgRNAs targeting downstream effectors of HLF such as LPXN or PXN, or targeting SMARCA4, also differentially distributed in metastases vs primary tumors?

7. Was the library sequenced deep enough? Extended Figure 1a show very little reads per sgRNAs.. or am I understanding this plot wrong?

8. Analysis: How did authors determine the essentiality score? The essentiality score is mostly used in the context of dropout screens that look for depleted sgRNAs. In this case, authors look for enriched sgRNAs in metastases, so this term is a bit confusing to me. How is the score computed and what is the significance threshold? Wouldn't logFC be a better metric for enrichment?

9. How did the authors incorporate information about biological replicate into the CRISPR screen analysis? How consistent are hits across mice and how were the results compared across mice? The paired test function of MAGECK can be used to find consistent results across replicates, considering primary and metastasis from the same mouse as a paired sample. Is Fig. 1a generated adding up all the mice? Can you plot enrichment of the chosen 10 genes per mouse in LogFC?

Additional questions:

1. Why switch cell line from UMRC2 and A498 for the screen to 786-O for the overexpression of the hits?
2. The authors find that HLF is downregulated in primary tumors of patients with increased chance of metastasis, is this specifically in the tumor thrombus? Are these cells the cells that go on to form metastases, and therefore secondary lesions have lower HLF levels than their paired primary tumors? These results suggest a heterogeneity in the primary tumor sample. Can authors analyze published human scRNAseq datasets (e.g. <https://www.nature.com/articles/s42003-024-06478-x>) to identify what subset of cells in primary tumors downregulate HLF and what is their transcriptional profile? The analysis could include LPXN and other factors that regulate or are regulated by HLF.

3. In Figure 2, authors perform an orthotopic injection of HFL depleted, WT or overexpressing 786-O cells. Did the tumors have different morphology? Was the invasive front different? Do the metastases look different?

4. The cell competition experiment in Fig 3I is very elegant. If the authors could perform this in vivo it would add a lot to the manuscript. It would also provide insight into the distribution of HLF low and high cells in primary tumors: are depleted cells enriched at the invasive front?

Reviewer #3

(Remarks to the Author)

Summary

It is now generally accepted that metastasis is not a result of novel mutations, yet remains the leading cause of cancer-associated deaths. There is therefore a great need to understand non-mutational routes to metastasis. The work here contributes to this, by elucidating a mechanism through which HLF is suppressed by SMARCA4, via enhancer regulation, during metastatic progression. The authors elucidate a detailed mechanism through which HLF contributes to metastasis through a combination of in vitro and in vivo assays, demonstrating that HLF loss downregulates LPXN which, via p-paxillin, enhances the integration of collagen stiffness to potentiate cell migration. Lastly, the authors extend their findings to gain clinical relevance by demonstrating the potential to modulate HLF expression via a SMARCA4 PROTAC, and through a panel of non-RCC cancer cell lines, demonstrate that their findings are relevant across multiple solid cancers.

The authors work relates to HLF. As the authors cite, this protein has previously been identified to promote multiple-organ distant metastases in lung cancer, where mechanisms related to metabolism, NF- κ B/p65 signalling and PPAR are presented. It has also been identified as part of a 4 gene signature for Lung Adenocarcinoma (Wu 2022). Its role in survival in other cancers has also been highlighted, such as in glioma (2021 Liu) and ovarian cancer progression (Han 2023).

The work is extensive, but does not bring any technical advances, so the significance appears to be restricted to the novel biological insights. The biological findings of this work do not shift our conceptual understanding of metastasis, often using reasoning that relies on our current assumptions regarding metastasis, such as the expected role of metastasis-associated

genes in migration and invasion in vitro that was used to narrow down which gene to further investigate. What is novel is the role of HLF specifically in RCC metastasis, and the mechanistic and regulatory insight presented in this work. The finding that the role of HLF in invasion and its regulation by SMARCA4 extends beyond ccRCC to other solid tumours poses an interesting potential common requirement across metastasis from solid tumours.

Comments on the writing

1. The terminology used in the writing is often imprecise and lacks clarity. For instance, line 123, 'genes ... inversely correlated with prognosis' would imply genes where high expression lead to worse prognosis, which is not what the authors are trying to say in this context. Similarly, in line 161, the authors use 'cell growth' where proliferation would have been more accurate. With regards to the HLF enhancer, the text states in line 357 "the HLF enhancer region". Is this a candidate/putative enhancer? Is there literature to confirm that it is a HLF enhancer region that requires citation? Otherwise to jump to claiming that an apparent enhancer in the region of HLF is the HLF enhancer region reads unsubstantiated.

2. With numerous publications providing various mechanisms behind the role of HLF in cancers, and the claims the authors make about their mechanism being relevant across multiple cancers, the authors could do more to place their work into the current literature and suggest how their findings compare with alternative mechanistic explanations, such as NF- κ B signalling or Hippo signalling for instance.

3. Some of the language could be toned down to better reflect the data. In line 210, the authors claim based on comparing migration in 5 cell lines a 'strong negative correlation'. Without quantifying a correlation coefficient. Additionally, at the very end of the discussion, the link from their work to circadian-regulated metastasis stretches quite far beyond the scope and topic of the present work and could be toned down.

Comments regarding experimental work and interpretation

4. The results of the in vivo CRISPR screen appear to conflict with what we would logically expect and some further explanation or plots to elaborate on this would be beneficial. Given the bottlenecks created by metastasis, and the fact that these derive from the primary tumours, it is surprising to see that, in many lung metastases, there is as much if not higher gRNA representation as in the primary tumours with no clear evidence of strong enrichment of particular gRNAs relative to the primary. The screen sgRNA coverage should therefore be better described. Additionally, to see the essentiality score plots for the primary sites, and how these overlap with the metastasis data, would also be beneficial to this manuscript. As the later results highlight that HLF is not impacting primary tumour growth, it would be beneficial to see that this is matched also in the initial screen data, and to see that the metastatic enrichments are indeed distinct from the primary would additionally add confidence to the screen.

5. The authors use genome-wide in vivo screening to identify candidate hits involved in metastatic progression. UBE4B is one of the strongest hits, is highlighted as one that is identified for further investigation, the text says "we then overexpressed these genes", yet UBE4B is not overexpressed in the figures. If there were any negative results here that lead to its exclusion, or any other reason to not move forward with this target, it would be worth mentioning or justifying.

6. The expression of the V5 tagged proteins is variable and often faint by western blot, and there is no quantification of endogenous levels. It is also notable that the only gene to have a strong phenotype upon overexpression is also by far the most strongly present in the western blot. While the subsequent work is not undermined by this, the approach of using the V5 overexpression cells in collective migration and single cell invasion assays as justification for focusing on HLF as a follow-up of an in vivo genome-wide genetic screen does not build the strongest case for the rationale of the study. Could the strong effect of HLF in comparison to the other proteins tested simply reflect the differences in expression levels?

7. The focus on in vivo metastasis necessitates a lot of in vivo work. While there are many in vivo experiments that robustly demonstrate the importance of HLF in metastasis in their models, much of the mechanistic work is carried out in vitro, including for instance the RNA sequencing that is used for identifying collagen binding as important. It would have been nice to see RNA sequencing, and the ChIP if possible, done with HLF KO in vivo on the metastatic nodules.

Minor comments re figure standards

8. The representative regions in figure 1E are not representative when considering the overall quantification.

9. There are some typos in the figures and inconsistencies in presentation e.g. use of hyphens and underscores, while in some figures the text within them is illegible.

10. Most of the bioluminescence images look to be oversaturated and take over most of the animal, which may call into question their quantification and separation from metastases.

References

Wu, Y.; Yang, L.; Zhang, L.; Zheng, X.; Xu, H.; Wang, K.; Weng, X. Identification of a Four-Gene Signature Associated with the Prognosis Prediction of Lung Adenocarcinoma Based on Integrated Bioinformatics Analysis. *Genes* 2022, 13, 238. <https://doi.org/10.3390/genes13020238>

Liu, QingLin MD; Ge, Huijian MD; Liu, Peng MD; Li, Youxiang * . High Hepatic leukemia factor expression indicates a favorable survival in glioma patients. *Medicine* 100(6):p e23980, February 12, 2021. | DOI: 10.1097/MD.00000000000023980

Han, T., Chen, T., Chen, L. et al. HLF promotes ovarian cancer progression and chemoresistance via regulating Hippo signaling pathway. *Cell Death Dis* 14, 606 (2023). <https://doi.org/10.1038/s41419-023-06076-5>

Reviewer #4

(Remarks to the Author)

Zhou et al. performed an in vivo genome wide CRISPR/Cas9 screen to identify drivers of lung metastasis in clear cell renal carcinoma (ccRCC). They identified hepatic leukemia factor (HLF) silencing as being both necessary and sufficient for lung metastasis. They further show that HLF modulates tumor cell interactions with the extracellular matrix, and that epigenetic silencing of HLF by the SWI/SNF ATPase, SMARCA4, drives lung cancer metastasis. Although the data presented in the manuscript are both extensive and rigorous, roles for HLF silencing in multiorgan metastasis are widely known. Linking SWI/SNF-mediated silencing to this process represents in significant advance. However, the SMARCA4 degrader AU-15330 studies are limited to tissue culture experiments. Therefore, the therapeutic implications of SWI/SNF inhibition are not well supported by the data presented. Providing in vivo pharmacologic or genetic evidence that SWI/SNF inhibition blocks metastasis via HLF would better support the major conclusion – that targeting the SMARCA4-HLF axis offers a therapeutic strategy for metastatic cancer.

Reviewer #5

(Remarks to the Author)

General comments for the author

This manuscript primarily elucidates the role of HLF as a tumor-suppressing regulator. Specifically, HLF modulates the expression of leupaxin, which serves as a mechanical signal for collagen, and effectively inhibits cancer cell migration and lung metastasis by facilitating the integration of pile protein with the actin cytoskeleton. Targeting the SMARCA4-HLF axis presents a promising therapeutic strategy. Indeed, this study is highly intriguing.

Major concern:

1. Through a series of in vivo and in vitro experiments, the author rigorously validated the functional role of HLF by employing knockdown, knockout, and overexpression techniques. Following this confirmation, the direct target genes of HLF, which mediate its impact on cell migration, were analyzed using ChIP-seq and RNA-seq data. To further verify the role of these target genes, the study explored how the HLF-LPXN-PXN axis coordinates cytoskeleton and collagen mechanical cues. The overall design of this research is well-conceived, employing a multifaceted verification approach with clear logical progression. Notably, the author conducted an in vivo CRISPR screen to identify suppressors of metastasis, yielding genes such as UBE4B, CYCS, HLF, FOSB, B4GALT6, and TRMT5 (Extended Data Fig. 1b-d). However, it remains unclear why the authors chose to focus on HLF for further investigation and whether the other screened genes have undergone revalidation or have been discarded.
2. HLF has been reported to be a rhythmically regulated gene. Does its expression in normal renal tissues or cells change rhythmically? If its expression changes, can we discuss the function of HLF in normal renal tissue and its role in the process of malignant transformation?
3. This study points out that the expression of HLF is regulated by multiple mechanisms, and proposes that SMARCA4 is the key upstream regulator of HLF. Can the author prove that the above results are indeed present in the occurrence and development of renal cancer through clinical data analysis?
4. This study indicates that the efficacy of the SMARCA4 degrader AU-15330 is diminished due to the absence of HLF, so could the authors discuss potential therapeutic strategies for HLF low-expressing cells? Does this treatment strategy have pan-cancer adaptability?
5. "Ethical approval" Document ID of all ethical approvals must be given in the manuscript.

Minor concern:

None.

Reviewer #6

(Remarks to the Author)

Version 1:

Reviewer comments:

Reviewer #1

(Remarks to the Author)

Reviewer #2

(Remarks to the Author)

The authors have addressed all of my criticisms.

However, they have not integrated their clarifications in the manuscript, which I think would be important for reproducibility of the study. In particular, the authors should take greater care in describing how they analyzed the screening results in the Methods section, including how they calculated the essentiality score and how they integrated the information from multiple biological replicates. I am not entirely convinced that picking "smallest adjusted p-values" as representatives is a good method to integrate information from multiple mice, and I would have like to see the screening results analyzed in an alternative way leading to the same top hits. This strategy should at least be carefully reported in the method section.

Extended Data Figure 9r and 9s and related text: "we observed that HLF and SMARCA4 exhibit a similar distribution in the UMAP analysis" sounds like over-interpretation of the UMAP, which is just a visualization of gene expression. The authors should subcluster the tumor cells if they want to state that HLF and SMARCA4 are expressed by the same subset of tumor cells. If the two genes are expressed by the same subsets of cells, how is it possible that they are inversely correlated? The authors should calculate actual correlation. Given the sparsity of the dataset, working with signatures rather than single genes would also increase confidence.

Reviewer #3

(Remarks to the Author)

The authors have addressed all my previous points, I have no further comments.

Reviewer #7

(Remarks to the Author)

In the manuscript by Zhou et al., the authors report the identification of HLF as a potent suppressor of lung metastasis through an in vivo genome-wide CRISPR screen, conducted following subcutaneous injections of clear cell renal cell carcinoma (ccRCC) cells into mice. They not only validated the role of HLF in suppressing lung metastasis using multiple orthogonal approaches, but also elucidated the underlying mechanisms—demonstrating that HLF regulates leupaxin expression, paxillin phosphorylation, actin cytoskeleton organization, and cellular migratory potential in collagen matrices. The authors further showed that HLF is epigenetically silenced in metastatic ccRCC, likely via SMARCA4-mediated repression, and that knockdown of SMARCA4 increases HLF expression and reduces lung metastasis in mice bearing orthotopically implanted ccRCC tumors. Lastly, they extended their findings beyond kidney cancer, showing that either HLF overexpression or pharmacological upregulation of HLF via AU-15330—a PROTAC targeting SMARCA4—reduced cell invasion across multiple cancer types.

Overall, this is a rigorous and comprehensive study supported by an impressive body of well-controlled experiments. The authors have satisfactorily addressed the major concerns raised by the reviewers, and the revised manuscript is compelling. I recommend this manuscript for publication in Nature Communications.

Reviewer #8

(Remarks to the Author)

No

REVIEWER COMMENTS

Reviewer #1 (Remarks to the Author): expert in ECM (collagen, stiffness)

Reviewer comments for NCOMMS-24-72086-T

“SWI/SNF ATPase - silenced HLF potentiates lung metastasis in solid cancers”

In this manuscript the authors use subcutaneous injection of clear cell renal cell carcinoma (ccRCC) cells and a genome wide CRISPR screen to identify a novel suppressor of metastasis – Hepatic leukemia factor or HLF. Of note the subcutaneous injections used for the genome wide CRISPR screen are less than ideal for spontaneous metastasis studies; however, the authors perform several validation experiments using orthotopic injection of ccRCC cells. The authors also continue by providing substantial mechanistic detail describing how HLF influences cell invasion by modifying the way in which cells integrate signals from the collagen extracellular matrix (ECM). HLF regulates levels of LPXN to inhibit cell adhesion signaling through paxillin and the actin cytoskeleton dynamics necessary for cell invasion. In addition, the authors determine how HLF is downregulated in metastatic cells by looking at HLF enhancer regions to discover that SMARCA4 is a major repressor of HLF transcription. These findings suggest that inhibition of SMARCA4 activity might be effective at limiting metastatic disease in ccRCC.

Overall, this is an extensive and comprehensive body of work that is experimentally well designed with robust data of high quality. Multiple cell lines, in vitro and in vivo models are used to support the manuscript’s main conclusions. Moreover, various methodologies are used to investigate molecular mechanisms including gain and loss of function studies as well as next generation sequencing based methods to explore genome-wide and HLF-specific regulation of gene expression (RNA-seq, ATAC-seq, CHIP-seq). Finally, the work is somewhat novel, and simultaneously holds great potential to provide broad clinical benefit for patients with various types of cancer using a SMARCA4 inhibitor (PROTAC-targeted degradation of SMARCA4) that has shown efficacy in targeting prostate cancer cells and that is currently in phase I clinical trials for metastatic uveal melanoma and advanced hematologic malignancies. For the above reasons, I recommend this article as suitable for publication in Nature Communications. However, I also include the following minor comments for consideration of their potential to add clarity and refinement to the current reported data.

Minor comments:

1. Previous reports cited in this manuscript (ref 12 and 13) indicate that HLF can activate c-JUN and inhibit p65-NFkappaB activity in hepatocellular carcinoma and non-small cell lung cancer respectively. Is there evidence for similar regulation of these factors in the ccRCC models and if yes, could their potential impact on metastasis be discussed?

Response: Thank you for the reviewer’s comments. We analyzed our RNA-seq data following HLF knockout and overexpression and found no changes in the p65-NFkB pathway, as neither NFKB1 nor RELA appeared in the differentially expressed gene (DEG) list. However, we observed altered expression of C-Jun, which decreased ($\log_2FC = -0.24$, $p_{adj} = 0.01$) upon HLF knockout

and increased ($\log_2FC = 1.57$, $padj = 3.6E-6$) upon HLF overexpression. C-Jun has been reported to promote metastasis in luminal breast cancer¹, gastric cancer² and hepatocellular carcinoma³. However, in ccRCC, C-Jun is positively regulated by HLF, which acts as a tumor suppressor in our study. It remains unclear whether C-Jun plays a role in regulating metastasis in this context, which warrants further investigation. We have incorporated this discussion into the revised manuscript as shown in the quotation below (marked in red in the revised manuscript).

“...Additionally, previous studies have shown that HLF downregulation promotes multi-organ distant metastases in non-small cell lung cancer via NF- κ B/p65 signaling⁴. However, in our study, we observed no significant changes in NF- κ B/p65 signaling upon HLF regulation. Moreover, while HLF has been reported to transactivate C-Jun to promote hepatocellular carcinoma⁵, and C-Jun itself has been implicated in promoting metastasis^{2,3}, we found that in ccRCC, C-Jun is positively regulated by HLF, which acts as a tumor suppressor in our study. Whether C-Jun plays a role in metastasis regulation in ccRCC remains unclear and warrants further investigation.”

2. Continuing from the comment above, recent reports suggest that HLF can activate YAP1 activity in ovarian cancer to promote cancer cell stemness, and induce levels of Tensin 1 in prostate cancer to modulate prostate cancer cell survival, invasion and metastasis. It would be informative to comment on these studies and any potential relevance to the current mechanism described for ccRCC, which involves adhesion mediated integration of mechanical cues from the ECM to the actin cytoskeleton.

Han T, Chen T, Chen L, Li K, Xiang D, Dou L, Li H, Gu Y. HLF promotes ovarian cancer progression and chemoresistance via regulating Hippo signaling pathway. *Cell Death Dis.* 2023 Sep 14;14(9):606. doi: 10.1038/s41419-023-06076-5. PMID: 37709768; PMCID: PMC10502110.

Hao Zhou, Fang Wang, Tensin 1 regulated by hepatic leukemia factor represses the progression of prostate cancer, *Mutagenesis*, Volume 38, Issue 6, November 2023, Pages 295–304, <https://doi.org/10.1093/mutage/gead027>

Response: Thank you for the reviewer’s suggestions. We analyzed our RNA-seq data following HLF knockout and overexpression found no changes in YAP1 expression. However, we observed significant alterations in tensin 1 (TNS1) expression, which increased ($\log_2FC = 0.89$, $padj = 3E-26$) upon HLF knockout and decreased ($\log_2FC = -1.88$, $padj = 1.9E-56$) upon HLF overexpression. We have incorporated this discussion into the revised manuscript as shown in the quotation below (marked in red in the revised manuscript).

“... the role of HLF in solid tumors was not appreciated until recently, including carcinogenesis in liver, distant metastases in non-small cell lung cancer and cell stemness in ovarian cancer¹²⁻¹⁴...”

“...In addition, given that HLF is a transcription factor, it likely regulates multiple genes and signaling pathways. We acknowledge that additional pathways may also play a role in HLF-mediated metastasis regulation in ccRCC and that HLF signaling pathways vary across different cancer types. For example, tensin1 (TNS1), a focal adhesion protein that regulates the molecular linkage between the extracellular matrix and the cytoskeletal network to control cell migration, has been reported to mediate HLF-driven metastasis suppression in prostate cancer⁵⁵. However, in

ccRCC, we found that HLF negatively regulates tensin1 expression, highlighting the cancer type-specific differences in HLF signaling.”

3. In some instances, images of the lung bioluminescence appear “cut off” and fail to show sufficient coverage of the mouse ventral midsection. Is there an explanation for this? Alternatively, it would be advisable to include some IHC (e.g. H&E) of lung metastases in addition to the bioluminescence imaging (in vivo and ex vivo) to support the robust quality of these data. Lung sections could also be used to further validate the proposed mechanism where appropriate.

Response: Thank you for the reviewer’s comments and suggestions. The “cut-off” appearance in the lung bioluminescence images, as noted by the reviewer, is due to the use of black cardboard to cover the primary kidney tumor. This was done to prevent signal interference from the primary tumors in the kidney, which emits a much stronger signal than lung metastasis. The image shown is a representative example to illustrate the relative signal compared to the primary tumor in the same mouse; however, we do not use this signal for lung metastasis quantification. Instead, after sacrificing the mice, we dissect the lungs and perform *ex vivo* bioluminescence imaging in the presence of the luciferase substrate, luciferin. This provides a more accurate signal, which we use for quantification. Since bioluminescence intensity is reported to correlate directly with tumor volume, this method is widely recognized and extensively used in lung metastasis studies^{6,7}. Additionally, we agree that performing H&E staining for lung metastasis is a valuable approach. We have conducted this experiment during the revision process. Please see the data below:

Extended Data Fig. 9v

4. While current data support a role for HLF in modulating collagen derived mechanical cues in driving ccRCC cell invasion (3D collagen and “2D stiff collagen” models), these data would be better supported by the culture of cells on a range of stiffnesses that correspond to normal renal ECM environments as well as soft and stiff disease states. For example, these studies could be accomplished using polyacrylamide hydrogels prepared with different crosslinker concentrations/stiffnesses followed by conjugation with collagen ECM. Such studies would also permit evaluation of the HLF-LPXN-Paxillin axis in different mechanical environments.

Response: Thank you for the reviewer’s suggestions. Regarding the stiffness of human kidney tissues, the previous publication showed that the Young’s modulus in ccRCC tumor tissue, tissue adjacent to the tumor biopsy site, and the cortex and medulla region of rejected transplant kidneys⁸. Their findings showed that ccRCC tumor tissues are less stiff than the healthy renal cortex (with mean Young’s modulus values of 3082 ± 963 Pa and 5480 ± 1409 Pa, respectively). No significant difference in stiffness was observed between the tumor and adjacent tissue (4303 ± 1300 Pa) or medulla tissue (4813 ± 1113 Pa). Unlike cancers such as breast and lung cancer,

where tumor tissues are significantly stiffer than normal tissues, kidney cancer exhibits similar stiffness to normal renal tissue.

The stiff collagen coating model we used, as referenced from a previous publication⁹, actually has a lower stiffness than that of real human tissues. To better mimic physiological conditions as the reviewer suggested, we performed cell migration tracking on surfaces with stiffness levels corresponding to real tissues (~3 kPa and ~6 kPa) using the crosslinker Genipin according to a publication¹⁰. As shown in the data below, under both collagen stiffness conditions, HLF knockout increased cell migration, as indicated by increased velocity, Euclidean distance, and directionality.

Figure legend: wind-rose plots showing cell tracks (a), migration velocity (b), Euclidean distance (c) and directionality (d) of the tracked 786-O cells transduced with sgCtrl or sgHLF and cultured on ~3kPa and ~6kPa collagen surface. n=22 and 23 tracked cells in sgCtrl and sgHLF with 3kPa collagen, n=25 tracked cells in sgCtrl and sgHLF with 6kPa collagen. One-way ANOVA followed by a post hoc Dunnett-t-test, exact P values are indicated.

In addition, we would like to clarify that our intention was not to suggest that HLF directly modulates collagen-derived mechanical cues to drive ccRCC cell invasion. Instead, our findings indicate that HLF's role in regulating cell invasion depends on collagen with a relatively stiff status or on its ability to sense stiff collagen cues. Our reasoning is as follows: we initially observed that HLF function is strongly associated with collagen involvement, and through mechanistic studies, we found that HLF downstream regulators are critical for sensing mechanical cues and adapting the cytoskeleton to promote migration, aligning with the stiff tumor microenvironment. Thus, our findings emphasize that HLF function depends on collagen (with our stiff collagen coating model based on collagen's role as the primary ECM component responsible for tissue stiffness), rather than HLF regulating or being regulated by collagen stiffness.

5. It would be nice to see evidence for differential LPXN localization at adhesion sites in cells with high vs. low levels of HLF. This would strengthen the mechanism of HLF induction of LPXN to limit Paxillin phosphorylation and adhesion signaling.

Response: Thank you for the reviewer's valuable suggestions. To examine LPXN localization at adhesion sites, we performed immunofluorescence staining for LPXN and the cytoskeletal marker F-actin in 786-O cells with and without HLF knockout. Our results showed a reduction in LPXN levels at adhesion sites upon HLF depletion, consistent with the decreased LPXN protein levels observed in western blot analysis and the increased paxillin phosphorylation at adhesion sites reported in our original manuscript. further support our conclusion that HLF loss reduces LPXN levels, leading to enhanced paxillin phosphorylation and ultimately promoting cell migration.

6. It is unclear whether the sample population size (n) is reported for TNMplot data, for example in Figure 1h.

Response: Thank you for the reviewer's comments. We have added the sample population size (N) for the TNMplot data in Figure 1h of the revised manuscript. Please see the revision below.

7. In Figure 4 and Extended Data Figure 5, it appears that ICAM2 and ADAMTSL1 have similar impacts on cell invasion as compared to AJAP1. An additional rationale for LPXN selection and elimination of these targets from subsequent analyses would be helpful.

Response: Thank you for the reviewer's comments. As noticed by the reviewer, the downstream targets of HLF, including ICAM2, ADAMTSL1, AJAP1, and LPXN, significantly inhibit cell invasion. This is expected, as HLF is a transcription factor, and these candidates were identified through RNA-seq and ChIP-seq analyses of HLF, followed by further validation. We acknowledge that each of these genes contributes to some extent to the phenotype mediated by HLF. While additional pathways may also be involved, our focus was on identifying and validating the most significant one. To achieve this, we assessed the phenotypic effects of individual gene overexpression and performed rescue experiments (HLF knockout with individual gene expression) using both transwell invasion assays and 3D collagen spheroid invasion assays. Based on these experiments, we selected AJAP1 and LPXN as the top two candidates for further validation. While both *AJAP1* and *LPXN* overexpression can inhibit cell invasion, we observed that while *AJAP1* knockdown had a mild inhibitory effect (Extended Data Fig. 6b), *LPXN* knockdown significantly promoted cell invasion (Extended Data Fig. 6d). This result aligns with the overexpression phenotype, ultimately identifying LPXN as the most critical HLF target mediating this effect. We have acknowledged the limitations of our study in the revised manuscript's Discussion section. Please see the revision shown in the quotation below (marked in red in the revised manuscript).

"...In addition, given that HLF is a transcription factor, it likely regulates multiple genes and signaling pathways. We acknowledge that additional pathways may also play a role in HLF-mediated metastasis regulation in ccRCC..."

8. Extended Data Figure 6f appears to be missing protein labels for the left panel of immunoblots.

Response: Thank you for the reviewer's comments. We're sorry for the oversight and have added corresponding labels to the immunoblots in Extended Data Figure 6f. Please also see the updated figure below:

Extended Data Fig. 6f

Reviewer #2 (Remarks to the Author): expert in In vivo CRISPR screens (metastasis)

In this manuscript, Zhou and colleagues perform an in vivo genome-wide CRISPR-Cas9 screen in ccRCC to identify factors that govern lung metastasis. They identify HLF as a novel tumor suppressor, whose silencing in primary tumor promotes invasion and metastasis. The authors performed a detailed and elegant molecular dissection of the mechanism of action of the HLF-LPXN-PXN axis in regulating cell-matrix adhesion and metastasis. Collectively, the authors have conducted a very thorough analysis of the role of HLF in regulating metastasis of ccRCC. I hope

that the following comments – mostly regarding the screen - will help authors improve their manuscript.

Comments and technical questions regarding the in vivo screen:

1. Setting: the authors perform a screen for lung metastasis by injecting two human ccRCC lines harboring a genome wide KO library subcutaneously into NSG mice. This setting is not ideal for a screen that aims at identifying factors that regulate metastasis because the molecular requirement for invasion and metastatic dissemination from the skin might differ from those from the kidney. Moreover, the use of NSG mice prevents analysis of the immune regulation of the metastatic cascade. While this might be advantageous to focus on non-immune effects, it is still a reductionistic view of metastasis. The authors should comment on why they chose the subcutaneous setting for the screening rather than the orthotopic model used for validation.

Response: We thank the reviewer for bringing up this important point. Indeed, setting up the CRISPR screen at the orthotopic site of ccRCC would be ideal for identifying key factors that regulate tumor metastasis. Unfortunately, intrarenal injections are commonly limited to less than 50 μ L of injection volume¹¹ due to the structural rigidity of renal parenchyma. This inevitably creates technical difficulty for performing a whole genome screen with sufficient coverage. Nevertheless, we are planning to set up a more focused, small-scale screen in the kidney of immunocompetent mice as the reviewer suggested, which we would like to include in a separate study.

2. Coverage at transduction: authors state that they transduced 1×10^8 cells with the library (1×10^5 sgRNAs). This would mean a 1000x coverage if 100% of the cells were transduced, which is not the case with a MOI of 3%. What was the transduction efficiency and therefore the actual coverage at the time of library transduction?

Response: Thank you for the reviewer's comments. We apologize for not articulating it clearly. While transducing viral particles to ccRCC cells, we took caution to make sure that the majority of infected cells received only one viral particle containing a single sgRNA per cell. To achieve this, we controlled the transduction efficiency to around 30%, equivalent to ~ 0.3 MOI according to the use guide of GeCKO library¹². This led to ~ 300 coverage ($1 \times 10^8 \times 30\% / 1 \times 10^5$) at the time of library transduction.

3. Coverage at engraftment: authors state that they injected 1.5×10^7 cells per flank for a coverage of 200x. However, $1.5 \times 10^7 / 1 \times 10^5$ is 150x. Did the authors inject mice in both flanks?

Response: Thank you for the reviewer's comments. Yes, we performed injections into both flanks of NSG mice and we have included this description in the METHOD section.

4. Both sgRNAs and Cas9 are expressed constitutively in cell lines upon transduction. Therefore, during the subsequent 1-week culture in vitro of the cells, sgRNA distribution will be altered compared to the viral library (e.g. sgRNAs targeting essential genes will be lost). The abundance of sgRNAs targeting genes that regulate expansion in culture (proliferation) will also be altered. Did authors sequence the library in cells prior to injection? This would help in normalizing the

results for these effects (by setting a baseline distribution) and would determine the sgRNA distribution and coverage of the library that was ultimately screened in vivo. Moreover, authors could compute depleted sgRNAs in metastases vs pre-injection and depleted sgRNAs in primary vs pre-injection to identify genes that are specifically needed for metastasis but not for primary tumor growth (see next point). In any case, authors should show depletion of sgRNAs targeting essential genes and distribution of control sgRNAs.

Response: We appreciate the reviewer's valuable suggestions, and we totally agree that the sgRNA distribution of CRISPR library was altered after the 1-week expansion of virally transduced cells on culture dish. This was an intentional experimental design, because essential genes like those involved in cell division or protein synthesis would be dropped out during this process¹³. 1 week expansion time is normally sufficient to eliminate these essential genes, so that they would not interfere with our metastasis-centered screening. Notably, these essential genes were similar to "core fitness genes" (~1,500) identified by a previous study, which provided a global view of genetic vulnerabilities of cancer cells using multiple CRISPR screens¹⁴. In addition, a recent study expanded upon previous work and identified multiple groups of tissue-specific fitness genes by CRISPR screens in 930 cancer cell lines¹⁵. We retrieved all kidney-specific fitness genes from the latter work and compared them with our screening results shown in Fig. 1b. We observed that only 10 out of 430 kidney-specific fitness genes were found in the hit list of our screening results, including ARPC4, CHCHD2, CTU2, FERMT2, LRRC37A3, MCTS1, NFE2L3, PTK2, PTPN23, and TFRC. Moreover, none of them were top ranked in the hit list and therefore excluded for subsequent analysis. These results confirmed that 1-week expansion time is sufficient to deplete most sgRNAs targeting essential genes in ccRCC cells. Although we did not sequence the CRISPR library in viral particles or pre-injected cells, these data provided an alternative support to the notion that our identified genes specifically regulate ccRCC metastasis rather than tumor growth.

5. In addition to altering library distribution in vitro, the effect of the induced perturbation is going to affect every subsequent step of the screen in vivo: from engraftment to tumor growth, to dissemination, survival in circulation, colonization and growth at secondary site. All these processes are being screened at the same time. Engraftment by itself causes a massive bottlenecking effect. Perturbations that impact engraftment and tumor growth will be depleted or enriched during the first weeks of tumor growth in vivo, prior to metastatic dissemination. After colonization, perturbation will be also impacting growth at secondary site. While this can only be avoided using an inducible system, some controls and additional analyses can be conducted to estimate the bottlenecking and skewing of the library. Did the authors sequence the sgRNA library in tumors also upon the engraftment period of this model but prior to metastatic dissemination? This would be the ideal baseline for comparison. Can the authors identify which sgRNAs are lost between plasmid or viral library and experimental endpoint?

Response: We appreciate the reviewer's constructive suggestions. We agree that CRISPR-mediated genetic manipulation would affect every step of metastasis after tumor engraftment, including primary tumor growth (step 1), cancer cell dissemination and anoikis resistance (step 2), as well as tumor colonization and re-growth at distal sites (step 3). Whereas inducible system is ideal for dissecting each metastatic step, we decided to focus on step 2 and 3 in the case of

ccRCC, because distal metastasis rather than primary tumor growth significantly shortened patient survival in this tumor type. Based on the statistical data from American Cancer Society, the 5-year survival rate for kidney cancer (~80% being ccRCC) patients with localized or regional tumors is >75%, in sharp comparison to the 5-year survival rate of <20% for patients with metastatic tumors (<https://www.cancer.org/cancer/types/kidney-cancer/detection-diagnosis-staging/survival-rates.html>). Therefore, target screen for step 2 or 3 may be more beneficial for improving the prognosis of ccRCC patients. To do this, we compared the sgRNA enrichment of metastatic tumors with simultaneously-formed primary tumors, to identify potential targets regulating step 2 or 3. If we set the comparison baseline to sgRNA enrichment in tumors upon the engraftment period, we may identify essential targets for step 1-3 including those regulating primary tumor growth, which is unlikely the focus of our study. We deeply apologize that we cannot identify depleted sgRNAs between viral library and experimental endpoint in this study, because we did not sequence the CRISPR library in viral particles. Nevertheless, we sincerely hope the reviewer agree that the comparison baseline we chose here may lead to more clinically relevant targets.

6. Authors compare sgRNAs depleted in metastases vs primary tumors and identify sgRNAs enriched in metastasis whose inferring genes are metastasis suppressors. Did the authors also find sgRNAs depleted in metastasis but not in primary tumors that would hint at genes required for metastasis? Were sgRNAs targeting downstream effectors of HLF such as LPXN or PXN, or targeting SMARCA4, also differentially distributed in metastases vs primary tumors?

Response: Thank you for the reviewer's comments. As the reviewer suggested, we compared our hit gene list with genes required for ccRCC metastasis. Based on a previous evolutionary study of matched primary and metastatic biopsies from 100 ccRCC patients¹⁶, most metastasis driver genes were mutated or deleted in metastatic lesions. However, a few chromosomal sites were amplified in metastatic ccRCC tumors, including 1q25.1, 2q14.3, 5q35.3, 7q22.3, 8q24.21, 12p11.21, and 20q13.33. We retrieved ~100 genes located at these sites via the OMIM database (<https://www.omim.org>), and compared them with our hit gene list. Not a single overlapping gene was found, further confirming that our identified genes are metastasis suppressors. For the sgRNAs that were depleted in metastases but not in primary tumors, as the reviewer mentioned, this can suggest that the gene is not metastasis-related, there is also a possibility that the gene may actually promote metastasis, as its loss inhibits lung metastasis. However, the likelihood of false positives is high in this context. Therefore, it is more rational to identify metastasis suppressors using CRISPR screening. If the goal is to identify genes that promote metastasis, an **overexpression screening** would be a more appropriate approach. In addition, we examined the distribution of sgRNAs targeting LPXN, PXN, and SMARCA4 in harvested primary and metastatic tumors. Although the fold change (FC) values of sgRNAs exhibit high variations among our tissue samples, we did observe a trending difference in the LPXN sgRNA distribution. The FC ratio of LPXN sgRNA in metastasis vs. primary tumors is 1.91 ± 0.75 (mean \pm standard error), comparable to 1.66 ± 0.34 in the case of HLF. However, we observed little differences in the sgRNA distribution of SMARCA4 and PXN, possibly because these two genes promote metastasis and negatively correlate with HLF and LPXN. Therefore, SMARCA4 and PXN are more likely picked up by dropout screens, which require significantly higher coverage than our enrichment screens.

7. Was the library sequenced deep enough? Extended Figure 1a show very little reads per sgRNAs.. or am I understanding this plot wrong?

Response: Thank you for the reviewer's comments. We agree with the reviewer that the log2 values of normalized reads per sgRNA (or in other words, sgRNA representation) were not high in our sequenced samples (averagely around 1). The average sequencing depth was >200 per sample, which should lead to sufficient sgRNA coverage. Therefore, the relatively low sgRNA representation was likely due to experimental design and application of human ccRCC cell lines. We mainly followed the screening protocol from a pioneer study focusing on lung metastasis using murine lung cancer cells¹⁷. Out of three metastatic samples they sequenced, two samples exhibited ~2 sgRNA representation on average (Fig. 2C in Cell 2015, PMID: 25748654). We speculated that isogenic murine xenografts were associated with rapid formation of primary tumors and metastatic lesions in mice, leading to significantly lower dropout rates of transduced sgRNAs. Consistent with this argument, a recent study using human prostate cell line 22Rv1 to probe bone and lung metastasis resulted in even lower sgRNA representation in their lung metastases than ours (Fig. S2B in Oncogene 2024, PMID: 38454137). Moreover, we noticed that A498 cells, which form ccRCC xenograft tumors suboptimally than UMRC2 cells, turned to have less sgRNA representation in both primary tumors and metastatic lesions (Extended Fig. 1a).

8. Analysis: How did authors determine the essentiality score? The essentiality score is mostly used in the context of dropout screens that look for depleted sgRNAs. In this case, authors look for enriched sgRNAs in metastases, so this term is a bit confusing to me. How is the score computed and what is the significance threshold? Wouldn't logFC be a better metric for enrichment?

Response: Thank you for the reviewer's comments. We apologize for the confusion and we agree with the reviewer that essentiality score is commonly used for dropout screens. We borrowed this concept by arbitrarily defining an "essentiality score" for our enrichment screens based on the adjusted p-values calculated by MAGeCK. We can change it to "enrichment score" if the reviewer think it's necessary. First, MAGeCK was applied to generate FC and p-values for sgRNA enrichments of lung or liver metastatic lesions relative to their corresponding primary tumors. We noticed that *in vivo* metastatic screens led to fairly high variations in the FC values (elaborated in our response to the next point), so we decided to use p-values instead of FC as the enrichment criteria. Specifically, we calculated the MAGeCK results of 5 replicated mice, and genes essential for regulating lung metastasis were identified if at least 4 replicates showing similar metastatic organotropism to the lung. The reason for including this step is that renal tumors metastasize to the lung (70%) much more frequently than to the liver (18%) and other sites¹⁸, therefore genes consistently regulate lung metastasis may be more pathologically relevant to ccRCC. As a result, our defined "essentiality scores" for these genes were calculated based on the negative logarithm of representative p-values specifically adjusted for lung enrichments. Finally, we ranked these genes according to their essentiality scores, and there was no pre-set significance threshold. Nevertheless, all the top ten genes have adjusted p-values <0.1, which are equivalent to essentiality scores >1.

9. How did the authors incorporate information about biological replicate into the CRISPR screen analysis? How consistent are hits across mice and how were the results compared across mice?

The paired test function of MAGECK can be used to find consistent results across replicates, considering primary and metastasis from the same mouse as a paired sample. Is Fig. 1a generated adding up all the mice? Can you plot enrichment of the chosen 10 genes per mouse in LogFC?

Response: We greatly thank the reviewer for pointing out this important issue. We did observe variations of our screening results across replicated mice, which were exemplified by the FC distribution of 10 chosen genes as shown in the figure below. To incorporate data from multiple biological replicates, we performed the following steps: 1. We utilized the paired test function of MAGECK as the reviewer mentioned, by comparing the sgRNA enrichment of metastatic tumors with primary tumors in the same mouse. 2. We chose to compare p-values instead of FC, because FC was prone to show relatively high variations. 3. Out of 5 mice, we only incorporated the data from 4 mice with higher preferences for lung metastasis, excluding one data point with less pathological relevance (as discussed in our response to the previous point). 4. Out of 5 mice, the smallest adjusted p-values were chosen as the representative score and subjected to the final ranking analysis. By doing so, we took a comprehensive consideration of all replicated data instead of simply adding them up. We were glad to see that the subsequent phenotypic results of our study validated this approach.

Additional questions:

1. Why switch cell line from UMRC2 and A498 for the screen to 786-O for the overexpression of the hits?

Response: Thank you for the reviewer's comments. UMRC2, A498, and 786-O cells are all commonly used ccRCC cell models, and while all of them can be utilized for *in vivo* studies, 786-O cells are generally considered more suitable for *in vitro* studies such as transwell invasion and 3D spheroid invasion assays. Therefore, 786-O cells are prioritized for further *in vitro* validation. Additionally, we would like to clarify that we have used multiple cell lines for functional validation, including 786-O, Caki-1, UMRC6, and A498. Furthermore, we incorporated PDX-derived cell lines, such as XP258 and XP374. Using multiple cell lines enhances the robustness of our study.

2. The authors find that HLF is downregulated in primary tumors of patients with increased chance of metastasis, is this specifically in the tumor thrombus? Are these cells the cells that go on to form metastases, and therefore secondary lesions have lower HLF levels than their paired primary

tumors? These results suggest a heterogeneity in the primary tumor sample. Can authors analyze published human scRNAseq datasets (e.g. <https://www.nature.com/articles/s42003-024-06478-x>) to identify what subset of cells in primary tumors downregulate HLF and what is their transcriptional profile? The analysis could include LPXN and other factors that regulate or are regulated by HLF.

Response: Thank you for the reviewer's comments and suggestions. While not all patients with tumor thrombus develop distant metastasis, we found it significant that those with tumor thrombus and low *HLF* expression are more likely to have distant metastasis. The reviewer's understanding is correct—primary tumors exhibit heterogeneity, and tumors with decreased *HLF* expression have increased chance of metastasis.

Following the reviewer's suggestion, we analyzed the expression of *SMARCA4*, *HLF*, and *LPXN* in the scRNA-seq dataset published by Zvirblyte et al. The dataset includes a total of 3,039 tumor cells, with the following number of cells expressing these genes: *SMARCA4* – 133 cells, *HLF* – 91 cells, and *LPXN* – 24 cells. Since the number of cells expressing *LPXN* is too low to provide statistically meaningful analysis, we focused on evaluating the correlation between *HLF* and its negative upstream regulator *SMARCA4*. First, we observed that *HLF* and *SMARCA4* exhibit a similar distribution in the UMAP analysis. We then classified tumor cells into two groups: *HLF*-high (*HLF* expression > 0) and *HLF*-low (*HLF* expression = 0). Our analysis revealed an inverse correlation, where *SMARCA4* expression is lower in *HLF*-expressing cells and vice versa. These findings from the scRNA-seq data further support the role of *SMARCA4* as a critical upstream regulator of *HLF* expression. We have included the correlation data of *SMARCA4* and *HLF* in the revised manuscript (**Extended Data Fig. 9r and 9s**), with the corresponding modifications highlighted in red.

3. In Figure 2, authors perform an orthotopic injection of HLF depleted, WT or overexpressing 786-O cells. Did the tumors have different morphology? Was the invasive front different? Do the metastases look different?

Response: Thank you for the reviewer's comments. Regarding the orthotopic injection data in Figure 2, as mentioned by the reviewer, we didn't perform specific staining to compare the morphology between different groups in PDX tumors. However, during the revision process, to

address this question, we conducted an orthotopic injection of wild-type (WT) and HLF-depleted (KO) 786-O cells. We then performed H&E staining on both kidney primary tumors and lung metastases to assess potential morphological differences between control and HLF-depleted cells. The results are shown below:

In primary tumors, both WT and KO tumors exhibit classical ccRCC morphological features. Regarding the invasive front, as mentioned in the next comment, unlike the *in vitro* 3D spheroid invasion assay, where cells can precisely disseminate from the edge of the spheroid, real tumors *in vivo* contain numerous blood vessels allowing cells to invade into circulation for metastasis. This complexity makes it technically challenging to evaluate the invasive front within mouse tumors. In lung metastases (Mets), the metastatic WT tumor displays a more defined boundary against the lung parenchyma, with tumor cells exhibiting nuclear pleomorphism and prominent nucleoli, while the metastatic KO tumor demonstrates a more infiltrative growth pattern into the lung tissue, with tumor cells appearing more loosely arranged and blending with the lung parenchyma. It is interesting that we observed morphological differences in metastatic tumors, suggesting that HLF may influence factors such as cell metabolism or cytoskeletal dynamics, which could contribute to variations in metastatic tumor morphology. Future studies investigating these mechanisms would be valuable.

4. The cell competition experiment in Fig 3I is very elegant. If the authors could perform this *in vivo* it would add a lot to the manuscript. It would also provide insight into the distribution of HLF low and high cells in primary tumors: are depleted cells enriched at the invasive front?

Response: Thank you for the reviewer's comments. We conducted an orthotopic injection of mixed 786-O sgCtrl (RFP-labeled) and sgHLF (GFP-labeled) cells into NSG mice. After eight weeks, we performed immunofluorescence imaging on both primary kidney tumor tissues and lung metastases. However, unlike the *in vitro* 3D spheroid invasion assay—where cells can be precisely disseminated from the edge of the spheroid—real tumors *in vivo* contain numerous blood vessels, allowing cells to invade into circulation for metastasis. This makes it technically challenging to evaluate the invasive front within the tumor in mouse. Nevertheless, our staining results revealed: 1) in primary kidney tumors, sgHLF cells tend to form larger localized spheroids rather than integrating evenly with sgCtrl cells, 2) in lung metastases, we observed a significantly stronger GFP signal, indicating a higher presence of sgHLF cells. These findings further support that HLF loss promotes ccRCC lung metastasis. We have included this data in the revised manuscript.

Fig. 3n

Reviewer #3 (Remarks to the Author): expert in ccRCC metastasis, epigenetics

Summary

It is now generally accepted that metastasis is not a result of novel mutations, yet remains the leading cause of cancer-associated deaths. There is therefore a great need to understand non-mutational routes to metastasis. The work here contributes to this, by elucidating a mechanism through which HLF is suppressed by SMARCA4, via enhancer regulation, during metastatic progression. The authors elucidate a detailed mechanism through which HLF contributes to metastasis through a combination of *in vitro* and *in vivo* assays, demonstrating that HLF loss downregulates LPXN which, via p-paxillin, enhances the integration of collagen stiffness to potentiate cell migration. Lastly, the authors extend their findings to gain clinical relevance by demonstrating the potential to modulate HLF expression via a SMARCA4 PROTAC, and through a panel of non-RCC cancer cell lines, demonstrate that their findings are relevant across multiple solid cancers.

The authors work relates to HLF. As the authors cite, this protein has previously been identified to promote multiple-organ distant metastases in lung cancer, where mechanisms related to metabolism, NF- κ B/p65 signalling and PPAR are presented. It has also been identified as part of

a 4 gene signature for Lung Adenocarcinoma (Wu 2022). Its role in survival in other cancers has also been highlighted, such as in glioma (2021 Liu) and ovarian cancer progression (Han 2023).

The work is extensive, but does not bring any technical advances, so the significance appears to be restricted to the novel biological insights. The biological findings of this work do not shift our conceptual understanding of metastasis, often using reasoning that relies on our current assumptions regarding metastasis, such as the expected role of metastasis-associated genes in migration and invasion in vitro that was used to narrow down which gene to further investigate. What is novel is the role of HLF specifically in RCC metastasis, and the mechanistic and regulatory insight presented in this work. The finding that the role of HLF in invasion and its regulation by SMARCA4 extends beyond ccRCC to other solid tumours poses and interesting potential common requirement across metastasis from solid tumours.

Comments on the writing

1. The terminology used in the writing is often imprecise and lacks clarity. For instance, line 123, 'genes ... inversely correlated with prognosis' would imply genes where high expression lead to worse prognosis, which is not what the authors are trying to say in this context. Similarly, in line 161, the authors use 'cell growth' where proliferation would have been more accurate. With regards to the HLF enhancer, the text states in line 357 "the HLF enhancer region". Is this a candidate/putative enhancer? Is there literature to confirm that it is a HLF enhancer region that requires citation? Otherwise to jump to claiming that an apparent enhancer in the region of HLF is the HLF enhancer region reads unsubstantiated.

Response: Thank you for the reviewer's comments regarding writing clarity. For line 123, we have revised the inaccurate phrasing to "Genes... and exhibited lower expression levels associated with worse prognosis were selected for further evaluation". For line 161, we have replaced "cell growth" with "cell proliferation" and have applied this modification consistently throughout the revised manuscript. Regarding the statement on the "HLF enhancer", we apologize for the inaccurate description. We initially referred to this region as an enhancer because it contains both H3K27ac and H3K4me1 binding, which are markers of active enhancers¹⁹. However, instead of "HLF enhancer", the correct term should be "enhancer region that HLF binds to". We have revised this phrasing in the manuscript accordingly. All writing modifications are marked in red in the revised manuscript.

2. With numerous publications providing various mechanisms behind the role of HLF in cancers, and the claims the authors make about their mechanism being relevant across multiple cancers, the authors could do more to place their work into the current literature and suggest how their findings compare with alternative mechanistic explanations, such as NF- κ B signalling or Hippo signalling for instance.

Response: Thank you for the reviewer's suggestions. To investigate whether HLF-related pathways reported in other cancers are also present in ccRCC, we analyzed our RNA-seq data following HLF knockout and overexpression, focusing on these signaling pathways. Our analysis revealed no changes in the p65-NF- κ B pathway or Hippo signaling, as neither NFKB1, RELA

(p65-NF-κB pathway) nor YAP1, TAFAZZIN (Hippo pathway) appeared in the differentially expressed gene (DEG) list. However, we observed altered expression of C-Jun, which decreased ($\log_2FC = -0.24$, $padj = 0.01$) upon HLF knockout and increased ($\log_2FC = 1.57$, $padj = 3.6E-6$) upon HLF overexpression. C-Jun has been reported to promote metastasis in luminal breast cancer¹, gastric cancer² and hepatocellular carcinoma³. However, in ccRCC, while C-Jun is positively regulated by HLF, it functions as a metastasis suppressor, which contrasts with its role in other cancers, including hepatocellular carcinoma. We also observed significant alterations in tensin 1 (TNS1) expression, which increased ($\log_2FC = 0.89$, $padj = 3E-26$) upon HLF knockout and decreased ($\log_2FC = -1.88$, $padj = 1.9E-56$) upon HLF overexpression. We have incorporated this discussion into the revised manuscript as shown in the quotation below (marked in red in the revised manuscript).

“... the role of HLF in solid tumors was not appreciated until recently, including carcinogenesis in liver, distant metastases in non-small cell lung cancer and cell stemness in ovarian cancer¹²⁻¹⁴...”

“...In addition, given that HLF is a transcription factor, it likely regulates multiple genes and signaling pathways. We acknowledge that additional pathways may also play a role in HLF-mediated metastasis regulation in ccRCC and that HLF signaling pathways vary across different cancer types. For example, tensin1 (TNS1), a focal adhesion protein that regulates the molecular linkage between the extracellular matrix and the cytoskeletal network to control cell migration, has been reported to mediate HLF-driven metastasis suppression in prostate cancer⁵⁵. However, in ccRCC, we found that HLF negatively regulates tensin1 expression, highlighting the cancer type-specific differences in HLF signaling. Additionally, previous studies have shown that HLF downregulation promotes multi-organ distant metastases in non-small cell lung cancer via NF-κB/p65 signaling¹³. However, in our study, we observed no significant changes in NF-κB/p65 signaling upon HLF regulation. Moreover, while HLF has been reported to transactivate C-Jun to promote hepatocellular carcinoma¹², and C-Jun itself has been implicated in promoting metastasis^{53,54}, we found that in ccRCC, C-Jun is positively regulated by HLF, which acts as a tumor suppressor in our study. Whether C-Jun plays a role in metastasis regulation in ccRCC remains unclear and warrants further investigation.”

3. Some of the language could be toned down to better reflect the data. In line 210, the authors claim based on comparing migration in 5 cell lines a ‘strong negative correlation’. Without quantifying a correlation coefficient. Additionally, at the very end of the discussion, the link from their work to circadian-regulated metastasis stretches quite far beyond the scope and topic of the present work and could be toned down.

Response: Thank you for the reviewer’s comments. For line 210, we have adjusted our description of the correlation between *HLF* levels and migration ability in collagen in ccRCC cell lines by removing the word "strong" from the phrase “...a strong negative correlation...”. Regarding the discussion on linking our study to circadian-regulated metastasis, we agree that this topic extends beyond the scope of our present work. Therefore, we have removed this section from the revised manuscript. All modifications are marked in red in the revised manuscript.

Comments regarding experimental work and interpretation

4. The results of the in vivo CRISPR screen appear to conflict with what we would logically expect and some further explanation or plots to elaborate on this would be beneficial. Given the bottlenecks created by metastasis, and the fact that these derive from the primary tumours, it is surprising to see that, in many lung metastases, there is as much if not higher gRNA representation as in the primary tumours with no clear evidence of strong enrichment of particular gRNAs relative to the primary. The screen sgRNA coverage should therefore be better described. Additionally, to see the essentiality score plots for the primary sites, and how these overlap with the metastasis data, would also be beneficial to this manuscript. As the later results highlight that HLF is not impacting primary tumour growth, it would be beneficial to see that this is matched also in the initial screen data, and to see that the metastatic enrichments are indeed distinct from the primary would additionally add confidence to the screen.

Response: We greatly thank the reviewer for pointing out this important issue. We also noticed that the log₂ values of normalized reads per sgRNA (sgRNA representation) were not high in our metastatic samples. This was likely due to the intrinsic property of our experimental design. We mainly followed the screening protocol from a pioneer study focusing on lung metastasis (Cell 2015, PMID: 25748654). In order to collect all metastatic lesions from a sacrificed mouse, the whole lung tissues were homogenized and lysed for DNA extraction. Therefore, the incorporated sgRNAs were diluted in a large amount of background DNA from normal lung tissues. This inevitably created difficulties for the following sgRNA enrichment and sequencing step, no matter how deep the samples were sequenced. Similar to our results, there were less sgRNA representations in their metastatic lesions compared to primary tumors (Fig. 2C in Cell 2015, PMID: 25748654). Moreover, a recent study to probe bone and lung metastasis in a prostate cancer mouse model resulted in even lower sgRNA representation in their lung metastases (Fig. S2B in Oncogene 2024, PMID: 38454137). These paralleled studies collectively demonstrated that higher sgRNA representation in metastasis may not be feasible using this approach.

In terms of sgRNA coverage, we controlled the viral transduction efficiency to around 30%, equivalent to ~0.3 MOI according to the use guide of GeCKO library (Science 2014, PMID: 24336571). This led to an approximate coverage of 300, which was sufficient for a regular CRISPR screen. We deeply apologize that we cannot generate essentiality score plots for primary tumors, or examine how these overlap with our metastasis data, because we did not sequence the CRISPR library in viral particles or virally infected cells as the base line. The main reason was that distal metastasis rather than primary tumor growth significantly shortened patient survival in ccRCC. Based on the statistical data from American Cancer Society, the 5-year survival rate for kidney cancer (~80% being ccRCC) patients with localized or regional tumors is >75%, in sharp comparison to the 5-year survival rate of <20% for patients with metastatic tumors (<https://www.cancer.org/cancer/types/kidney-cancer/detection-diagnosis-staging/survival-rates.html>). We therefore planned to focus on targets specifically regulating ccRCC metastasis. Therefore, we utilized the paired test function of MAGECK software and directly compared the sgRNA enrichment of metastatic tumors with matched primary tumors in the same mouse. By doing so, we likely identified potential targets regulating tumor metastasis but not tumor growth. In other words, the metastatic enrichment of our screened-out genes had to be distinct from the primary, and we sincerely hope the reviewer agree that this approach may lead to more clinically relevant targets.

5. The authors use genome-wide in vivo screening to identify candidate hits involved in metastatic progression. UBE4B is one of the strongest hits, is highlighted as one that is identified for further investigation, the text says “we then overexpressed these genes”, yet UBE4B is not overexpressed in the figures. If there were any negative results here that lead to its exclusion, or any other reason to not move forward with this target, it would be worth mentioning or justifying.

Response: Thank you for the reviewer’s comments. We apologize for the oversight in explaining the exclusion of UBE4B. In fact, we tested multiple UBE4B constructs in different vector backbones but were unable to achieve overexpression (absence of Western blot bands and even a decrease in mRNA levels following transfection). Due to these challenges, we excluded UBE4B from further validation as a potential candidate. We have clarified this in the revised manuscript. Please also see the revision as shown in the quotation below (marked in red in the revised manuscript).

“We then overexpressed these genes in 786-O ccRCC cells, except for UBE4B for which we were unable to achieve exogenous expression ...”

6. The expression of the V5 tagged proteins is variable and often faint by western blot, and there is no quantification of endogenous levels. It is also notable that the only gene to have a strong phenotype upon overexpression is also by far the most strongly present in the western blot. While the subsequent work is not undermined by this, the approach of using the V5 overexpression cells in collective migration and single cell invasion assays as justification for focusing on HLF as a follow-up of an in vivo genome-wide genetic screen does not build the strongest case for the rationale of the study. Could the strong effect of HLF in comparison to the other proteins tested simply reflect the differences in expression levels?

Response: Thank you for the reviewer’s comments. We acknowledge the limitations of using V5-tagged plasmids to validate phenotypes via exogenous expression of CRISPR screening targets. However, since this serves as a secondary mini-screening to identify targets with the most robust phenotypic effects before conducting more refined validation experiments, this approach meets our preliminary screening needs. Regarding uneven expression levels, we attempted to optimize viral titers, lowering the titer of HLF while relatively increasing the titer of other targets. Despite this adjustment, the exogenous protein levels of targets other than HLF remained low. To address this, we treated the cells with the proteasome inhibitor MG132 before western blot analysis, which significantly increased B4GALT6, FOSB, and CYCS protein levels, bringing most of them in parallel with HLF (**Extended Data Fig. 1f**). This suggests that their instability contributed to the low levels detected in the original manuscript. Additionally, we repeated the transwell invasion assay using newly generated cell lines. Consistent with our original findings, HLF emerged as the most robust target, effectively inhibiting cell invasion (**Fig. 1d, Extended Data Fig. 1g**). We have also updated the revised version with the new data that reflects paralleled exogenous expression levels.

Extended Data Fig. 1f

Extended Data Fig. 1g

Fig. 1d

7. The focus on in vivo metastasis necessitates a lot of in vivo work. While there are many in vivo experiments that robustly demonstrate the importance of HLF in metastasis in their models, much of the mechanistic work is carried out in vitro, including for instance the RNA sequencing that is used for identifying collagen binding as important. It would have been nice to see RNA sequencing, and the ChIP if possible, done with HLF KO in vivo on the metastatic nodules.

Response: Thank you for the reviewer's suggestions. While conducting mechanistic studies *in vivo* is a valuable approach, we chose to use *in vitro* models for the following reasons: 1) precise experimental control: *in vitro* models allow precise control over experimental conditions, such as gene editing, whereas *in vivo* models involve complex physiological interactions that may interfere with targeted manipulations, making it challenging to focus on specific mechanisms; 2) reproducibility and consistency: *in vitro* experiments provide greater consistency and reproducibility, as conditions can be standardized. In contrast, *in vivo* models exhibit biological variability due to genetic, environmental, and physiological differences among animals.

Minor comments re figure standards

8. The representative regions in figure 1E are not representative when considering the overall quantification.

Response: Thank you for the reviewer's comments. Figure 1e presents representative clinical patient data. Due to individual heterogeneity among patients, HLF staining exhibits substantial variance, which can be observed in the quantification column. To better illustrate this variability, we have switched to a dot plot format, where each dot represents an individual value. The selected representative image displays the most robust paired data available.

9. There are some typos in the figures and inconsistencies in presentation e.g. use of hyphens and underscores, while in some figures the text within them is illegible.

Response: Thank you for the reviewer's comments. We have thoroughly reviewed the manuscript and figures, making the necessary modifications to ensure consistency in the presentation of the same elements. For example, the labeling in Extended Data Fig. 1i and 1j has been adjusted to match Fig. 1e.

10. Most of the bioluminescence images look to be oversaturated and take over most of the animal, which may call into question their quantification and separation from metastases.

Response: Thank you for the reviewer's comments. The representative bioluminescence images are intended to visually compare the signal intensity between the control and experimental groups. To maintain a consistent scale bar across different groups and between primary tumors and corresponding lung metastases, some mice with stronger signals may appear oversaturated. However, this does not affect quantification, as the observed saturation is solely due to the applied scale bar. During imaging, we strictly controlled the exposure time to prevent overexposure. Regarding the separation of metastases mentioned by the reviewer, this is due to the use of black cardboard to cover the kidney tumor. This was done to prevent signal interference from the primary tumors in the kidney, which emits a much stronger signal than lung metastasis. The image shown is a representative example to illustrate the relative signal compared to the primary tumor in the same mouse; however, we do not use this signal for lung metastasis quantification. Instead, after sacrificing the mice, we dissect the lungs and perform *ex vivo* bioluminescence imaging in the presence of the luciferase substrate, luciferin. This provides a more accurate signal, which we use for quantification. Since bioluminescence intensity is reported to correlate directly with tumor volume, this method is widely recognized and extensively used in lung metastasis studies^{6,7}.

References

Wu, Y.; Yang, L.; Zhang, L.; Zheng, X.; Xu, H.; Wang, K.; Weng, X. Identification of a Four-Gene Signature Associated with the Prognosis Prediction of Lung Adenocarcinoma Based on Integrated Bioinformatics Analysis. *Genes* 2022, 13, 238.

<https://doi.org/10.3390/genes13020238>

Liu, QingLin MD; Ge, Huijian MD; Liu, Peng MD; Li, Youxiang*. High Hepatic leukemia factor expression indicates a favorable survival in glioma patients. *Medicine* 100(6):p e23980, February 12, 2021. | DOI: 10.1097/MD.00000000000023980

Han, T., Chen, T., Chen, L. et al. HLF promotes ovarian cancer progression and chemoresistance via regulating Hippo signaling pathway. *Cell Death Dis* 14, 606 (2023).

<https://doi.org/10.1038/s41419-023-06076-5>

Reviewer #4 (Remarks to the Author): expert in Epigenetics, SWI/SNF, ChIP-seq and ATAC-Seq

Zhou et al. performed an *in vivo* genome wide CRISPR/Cas9 screen to identify drivers of lung metastasis in clear cell renal carcinoma (ccRCC). They identified hepatic leukemia factor (HLF) silencing as being both necessary and sufficient for lung metastasis. They further show that HLF modulates tumor cell interactions with the extracellular matrix, and that epigenetic silencing of HLF by the SWI/SNF ATPase, SMARCA4, drives lung cancer metastasis. Although the data presented in the manuscript are both extensive and rigorous, roles for HLF silencing in multiorgan

metastasis are widely known. Linking SWI/SNF-mediated silencing to this process represents in significant advance. However, the SMARCA4 degrader AU-15330 studies are limited to tissue culture experiments. Therefore, the therapeutic implications of SWI/SNF inhibition are not well supported by the data presented. Providing in vivo pharmacologic or genetic evidence that SWI/SNF inhibition blocks metastasis via HLF would better support the major conclusion – that targeting the SMARCA4-HLF axis offers a therapeutic strategy for metastatic cancer.

Response: Thank you for the reviewer’s suggestions. As AU-15330 is a VHL-based degrader, while most of ccRCC cell lines including 786-O are VHL null, so we applied genetic method. Briefly, we tried to knock down SMARCA4 which will lead to increased intrinsic expression of HLF which expects to suppress metastatic potential. We then further knocked out HLF following SMARCA4 knockdown to assess whether the phenotype could be rescued. Please see the data below, SMARCA4 knockdown increased intrinsic HLF expression, as indicated by elevated Flag-tagged HLF levels, and further HLF knockout resulted in complete loss of HLF expression (**Extended Data Fig. 9t**). While no significant difference was observed in primary tumor growth (**Extended Data Fig. 9u**), SMARCA4 knockdown reduced lung metastasis and further HLF knockout significantly increased lung metastasis (**Fig. 6o and 6p**). These findings support that HLF functions as a metastasis suppressor in ccRCC, with SMARCA4 acting as a key upstream regulator.

Extended Data Fig. 9t

Extended Data Fig. 9u

Fig. 6o

Fig. 6p

Reviewer #5 (Remarks to the Author): expert in HLF

General comments for the author

This manuscript primarily elucidates the role of HLF as a tumor-suppressing regulator. Specifically, HLF modulates the expression of leupaxin, which serves as a mechanical signal for collagen, and effectively inhibits cancer cell migration and lung metastasis by facilitating the integration of pile protein with the actin cytoskeleton. Targeting the SMARCA4-HLF axis presents a promising therapeutic strategy. Indeed, this study is highly intriguing.

Major concern:

1. Through a series of *in vivo* and *in vitro* experiments, the author rigorously validated the functional role of HLF by employing knockdown, knockout, and overexpression techniques. Following this confirmation, the direct target genes of HLF, which mediate its impact on cell migration, were analyzed using ChIP-seq and RNA-seq data. To further verify the role of these target genes, the study explored how the HLF-LPXN-PXN axis coordinates cytoskeleton and collagen mechanical cues. The overall design of this research is well-conceived, employing a multifaceted verification approach with clear logical progression. Notably, the author conducted an *in vivo* CRISPR screen to identify suppressors of metastasis, yielding genes such as UBE4B, CYCS, HLF, FOSB, B4GALT6, and TRMT5 (Extended Data Fig. 1b-d). However, it remains unclear why the authors chose to focus on HLF for further investigation and whether the other screened genes have undergone revalidation or have been discarded.

Response: Thank you for the reviewer's comments. We conducted *in vitro* experiments, including collective cell migration and single-cell invasion assays, by overexpressing these genes in ccRCC cells to identify those with the most robust phenotypic effects. In addition to HLF, we observed that other genes, such as B4GALT6 and TRMT5, also demonstrated effects in the transwell invasion assay, which is expected since all of these genes were identified through *in vivo* CRISPR screening. We acknowledge that other targets may also play a role in invasion and migration. However, based on results from more than three independent experiments, including additional experiments conducted during the revision process, HLF consistently emerged as the top candidate with the most robust phenotype. Therefore, we selected HLF for further validation.

2. HLF has been reported to be a rhythmically regulated gene. Does its expression in normal renal tissues or cells change rhythmically? If its expression changes, can we discuss the function of HLF in normal renal tissue and its role in the process of malignant transformation?

Response: Thank you for the reviewer's comments. Investigating whether *HLF* expression in normal renal tissues or cells fluctuates rhythmically is indeed an interesting question. However, this is not the focus of our current project. As Reviewer #3 also mentioned, linking our study to circadian-regulated metastasis extends beyond the scope of this work. Therefore, we have

removed this part of discussion from the revised manuscript. Nonetheless, we are interested in exploring the relationship between HLF-mediated metastasis and circadian rhythms, and this will be a focus of our future research.

3. This study points out that the expression of HLF is regulated by multiple mechanisms, and proposes that SMARCA4 is the key upstream regulator of HLF. Can the author prove that the above results are indeed present in the occurrence and development of renal cancer through clinical data analysis?

Response: Thank you for the reviewer's suggestions. We detected the expression of SMARCA4 and HLF in paired normal and ccRCC tumor samples and performed a correlation analysis to further provide evidence that SMARCA4 serves as an important upstream regulator of HLF. As shown below, SMARCA4 protein levels were elevated in ccRCC tumor samples compared to normal tissues (**Extended Data Fig. 9n and 9o**). Due to the lack of a reliable HLF antibody, we analyzed HLF mRNA levels in these paired tissues and observed a decrease in HLF expression in ccRCC tumor samples (**Extended Data Fig. 9p**). Furthermore, Pearson's correlation analysis revealed a negative correlation between SMARCA4 and HLF levels ($r = -0.5844$, $p = 0.0068$, **Extended Data Fig. 9q**).

Extended Data Fig. 9n

Extended Data Fig. 9o

Extended Data Fig. 9p

Extended Data Fig. 9q

4. This study indicates that the efficacy of the SMARCA4 degrader AU-15330 is diminished due to the absence of HLF, so could the authors discuss potential therapeutic strategies for HLF low-expressing cells? Does this treatment strategy have pan-cancer adaptability?

Response: Thank you for the reviewer's comments. In our study, *HLF* loss promotes metastasis, while SMARCA4 acts as an upstream regulator that negatively regulates *HLF* expression. Treatment with the SMARCA4 degrader AU-15330 increases *HLF* levels and reduces the invasive ability of cancer cells across multiple cancer types, including ccRCC, colon cancer, and skin cancer, as shown in Fig. 7i. These cancer types already demonstrated lower *HLF* levels in metastatic tissues compared to primary tumors, as shown in Fig. 1h. Therefore, our findings suggest that SMARCA4 degrader AU-15330 may serve as a potential therapeutic strategy for cancers where decreased *HLF* levels are associated with metastasis.

5. "Ethical approval" Document ID of all ethical approvals must be given in the manuscript.

Response: Thank you for the reviewer's comments. We have included more information for ethical approvals in the "Study approval" section. Please see the details below: "All animal experiments were conducted in accordance with the National Institutes of Health (NIH) guidelines and were approved by the Institutional Animal Care and Use Committee (IACUC) in the University of Texas Southwestern Medical Center (Protocol # 2019-102794). The deidentified human tissues used for qPCR, RNAScope, and IHC assays were reviewed by the UT Southwestern Human Research Protection Program (HRPP), which determined that the analysis does not qualify as human subject research under 45 CFR 46.102 and therefore does not require IRB approval or oversight."

Minor concern:

None.

Reviewer #6 (Remarks to the Author): ECR, co-reviewed with Reviewer #3

- 1 Han, Y. *et al.* Targeting c-Jun Is a Potential Therapy for Luminal Breast Cancer Bone Metastasis. *Mol Cancer Res* **21**, 908-921, doi:10.1158/1541-7786.MCR-22-0695 (2023).
- 2 Peng, Y. *et al.* Direct regulation of FOXK1 by C-jun promotes proliferation, invasion and metastasis in gastric cancer cells. *Cell Death Dis* **7**, e2480, doi:10.1038/cddis.2016.225 (2016).
- 3 Liu, C. *et al.* c-Jun-dependent beta3GnT8 promotes tumorigenesis and metastasis of hepatocellular carcinoma by inducing CD147 glycosylation and altering N-

- glycan patterns. *Oncotarget* **9**, 18327-18340, doi:10.18632/oncotarget.24192 (2018).
- 4 Chen, J. *et al.* Downregulation of the circadian rhythm regulator HLF promotes multiple-organ distant metastases in non-small cell lung cancer through PPAR/NF-kappab signaling. *Cancer Lett* **482**, 56-71, doi:10.1016/j.canlet.2020.04.007 (2020).
- 5 Xiang, D. M. *et al.* Oncofetal HLF transactivates c-Jun to promote hepatocellular carcinoma development and sorafenib resistance. *Gut* **68**, 1858-1871, doi:10.1136/gutjnl-2018-317440 (2019).
- 6 Haider, M. T. *et al.* Comparison of ex vivo bioluminescence imaging, Alu-qPCR and histology for the quantification of spontaneous lung and bone metastases in subcutaneous xenograft mouse models. *Clin Exp Metastasis* **41**, 103-115, doi:10.1007/s10585-024-10268-4 (2024).
- 7 Madero-Visbal, R. A. *et al.* Bioluminescence imaging correlates with tumor progression in an orthotopic mouse model of lung cancer. *Surg Oncol* **21**, 23-29, doi:10.1016/j.suronc.2010.07.008 (2012).
- 8 Abbott, A. *et al.* Development of a mechanically matched silk scaffolded 3D clear cell renal cell carcinoma model. *Mater Sci Eng C Mater Biol Appl* **126**, 112141, doi:10.1016/j.msec.2021.112141 (2021).
- 9 Park, J. S. *et al.* Mechanical regulation of glycolysis via cytoskeleton architecture. *Nature* **578**, 621-626, doi:10.1038/s41586-020-1998-1 (2020).
- 10 Ishihara, S., Kurosawa, H. & Haga, H. Stiffness-Modulation of Collagen Gels by Genipin-Crosslinking for Cell Culture. *Gels-Basel* **9**, doi:ARTN 148
10.3390/gels9020148 (2023).
- 11 Nishida, J., Miyakuni, K., Miyazono, K. & Ehata, S. An in vivo orthotopic serial passaging model for a metastatic renal cancer study. *STAR Protoc* **3**, 101306, doi:10.1016/j.xpro.2022.101306 (2022).
- 12 Shalem, O. *et al.* Genome-scale CRISPR-Cas9 knockout screening in human cells. *Science* **343**, 84-87, doi:10.1126/science.1247005 (2014).
- 13 Masoudi, M. Lessons from a genome-wide CRISPR-Cas9 screening: what researchers should know before start. *EXCLI J* **20**, 1615-1620, doi:10.17179/excli2021-4412 (2021).
- 14 Hart, T. *et al.* High-Resolution CRISPR Screens Reveal Fitness Genes and Genotype-Specific Cancer Liabilities. *Cell* **163**, 1515-1526, doi:10.1016/j.cell.2015.11.015 (2015).
- 15 Pacini, C. *et al.* A comprehensive clinically informed map of dependencies in cancer cells and framework for target prioritization. *Cancer Cell* **42**, 301-316 e309, doi:10.1016/j.ccell.2023.12.016 (2024).
- 16 Turajlic, S. *et al.* Tracking Cancer Evolution Reveals Constrained Routes to Metastases: TRACERx Renal. *Cell* **173**, 581-594 e512, doi:10.1016/j.cell.2018.03.057 (2018).
- 17 Chen, S. *et al.* Genome-wide CRISPR screen in a mouse model of tumor growth and metastasis. *Cell* **160**, 1246-1260, doi:10.1016/j.cell.2015.02.038 (2015).
- 18 Dudani, S. *et al.* Evaluation of Clear Cell, Papillary, and Chromophobe Renal Cell Carcinoma Metastasis Sites and Association With Survival. *JAMA Netw Open* **4**, e2021869, doi:10.1001/jamanetworkopen.2020.21869 (2021).

- 19 Creyghton, M. P. *et al.* Histone H3K27ac separates active from poised enhancers and predicts developmental state. *Proc Natl Acad Sci U S A* **107**, 21931-21936, doi:10.1073/pnas.1016071107 (2010).

REVIEWER COMMENTS

Reviewer #2 (Remarks to the Author):

The authors have addressed all of my criticisms.

However, they have not integrated their clarifications in the manuscript, which I think would be important for reproducibility of the study. In particular, the authors should take greater care in describing how they analyzed the screening results in the Methods section, including how they calculated the essentiality score and how they integrated the information from multiple biological replicates. I am not entirely convinced that picking "smallest adjusted p-values" as representatives is a good method to integrate information from multiple mice, and I would have like to see the screening results analyzed in an alternative way leading to the same top hits. This strategy should at least be carefully reported in the method section.

Response: Thank you for the reviewer's comments. We have incorporated detailed information on the analysis of the CRISPR screening results-including essentiality score calculation and integration from multiple biological replicates-into the Methods section of the revised manuscript.

In addition, in the original manuscript, the smallest negative p -values (representing the highest possible enrichments for our genome-wide dropout screens) out of 5 biological replicates were subjected to final ranking analysis. As the referee suggested, we have now performed an alternative analysis by incorporating negative p -values from all biological replicates, after application of the following data filtering criteria: the negative rank ("neg|rank" within the MAGeCK report) for lung metastases is higher than that of pairing liver metastases, and the positive rank ("pos|rank" within the MAGeCK report) for lung metastases is simultaneously lower than that of pairing liver metastases. These criteria were included to identify genes consistently regulating lung but not liver metastasis in all biological replicates, given that renal tumors metastasize to the lung (70%) much more frequently than to the liver (18%)¹ as we described in our previous response letter. Based on this alternative analysis, *HLF* was comprehensively ranked at #3 out of all screened genes in UMARC2 and A498 renal cancer cell lines. The above strategy has been added to the method section.

All modifications have been highlighted in red in the revised manuscript.

Extended Data Figure 9r and 9s and related text: "we observed that *HLF* and *SMARCA4* exhibit a similar distribution in the UMAP analysis" sounds like over-interpretation of the UMAP, which is just a visualization of gene expression. The authors should subcluster the tumor cells if they want to state that *HLF* and *SMARCA4* are expressed by the same subset of tumor cells. If the two genes are expressed by the same subsets of cells, how is it possible that they are inversely correlated? The authors should calculate actual correlation. Given the sparsity of the dataset, working with signatures rather than single genes would also increase confidence.

Response: Thank you for the reviewer's comments. In our manuscript, we observed a negative correlation between *SMARCA4* and *HLF* expression at the mRNA and protein levels (Supplementary Fig. 9n–q). Although these two genes are negatively correlated, they can still be

co-expressed in the same subsets of cells to varying degrees; their expression is not strictly mutually exclusive. However, in the single-cell dataset we analyzed during the first round of revision, the number of cells co-expressing both SMARCA4 and HLF was too limited to perform a reliable correlation analysis. We attempted to assess the correlation using the available single-cell data-including the small numbers of SMARCA4⁺/HLF⁺, SMARCA4⁺/HLF⁻, and SMARCA4⁻/HLF⁺ cells-and obtained an R² value of 0.5066 (see below).

However, due to limited number of cells and low expression levels, the reliability of this result cannot be ensured. Therefore, to maintain scientific rigor, we have removed the single-cell data and the related discussion from the revised manuscript.

References

- 1 Dudani, S. *et al.* Evaluation of Clear Cell, Papillary, and Chromophobe Renal Cell Carcinoma Metastasis Sites and Association With Survival. *JAMA Netw Open* 4, e2021869, doi:10.1001/jamanetworkopen.2020.21869 (2021).